# Sterol 14-α-demethylase is vital for mitochondrial functions and stress tolerance in *Leishmania major*

**Sumit Mukherjee**[¤a], **Samrat Moitra**[iD], **Wei Xu**[¤b], **Veronica Hernandez**, **Kai Zhang**[iD]*

Department of Biological Sciences, Texas Tech University, Lubbock, Texas, United States of America

¤a Current address: Division of Infectious Diseases, Department of Medicine, Washington University School of Medicine, St. Louis, Missouri, United States of America
¤b Current address: Department of Molecular Microbiology, Washington University School of Medicine, St. Louis, Missouri, United States of America
* kai.zhang@ttu.edu

**Data Availability Statement:** All relevant data are within the manuscript and its Supporting Information files.

**Funding:** This work was supported by a grant from the National Institutes of Health (AI099380 for KZ).

## Abstract

Sterol 14-α-demethylase (C14DM) is a key enzyme in the biosynthesis of sterols and the primary target of azoles. In *Leishmania major*, genetic or chemical inactivation of C14DM leads to accumulation of 14-methylated sterol intermediates and profound plasma membrane abnormalities including increased fluidity and failure to maintain ordered membrane microdomains. These defects likely contribute to the hypersensitivity to heat and severely reduced virulence displayed by the C14DM-null mutants (*c14dm⁻*). In addition to plasma membrane, sterols are present in intracellular organelles. In this study, we investigated the impact of C14DM ablation on mitochondria. Our results demonstrate that *c14dm⁻* mutants have significantly higher mitochondrial membrane potential than wild type parasites. Such high potential leads to the buildup of reactive oxygen species in the mitochondria, especially under nutrient-limiting conditions. Consistent with these mitochondrial alterations, *c14dm⁻* mutants show impairment in respiration and are heavily dependent on glucose uptake and glycolysis to generate energy. Consequently, these mutants are extremely sensitive to glucose deprivation and such vulnerability can be rescued through the supplementation of glucose or glycerol. In addition, the accumulation of oxidants may also contribute to the heat sensitivity exhibited by *c14dm⁻*. Finally, genetic or chemical ablation of C14DM causes increased susceptibility to pentamidine, an antimicrobial agent with activity against trypanosomatids. In summary, our investigation reveals that alteration of sterol synthesis can negatively affect multiple cellular processes in *Leishmania* parasites and make them vulnerable to clinically relevant stress conditions.

## Author summary

Sterols are well recognized for their stabilizing effects on the plasma membrane, but their functions in intracellular organelles are under explored, which hampers the development of sterol synthesis inhibitors as drugs. Our previous studies have demonstrated significant

The funder had no role in study design, data collection and analysis, decision to publish, or preparation of the manuscript.

**Competing interests:** NO authors have competing interests.

plasma membrane instability in the sterol biosynthetic mutant *c14dm⁻* in *Leishmania major*, a pathogenic protozoan responsible for cutaneous leishmaniasis causing 1–1.5 million infections a year. While the plasma membrane defects have undoubtedly contributed to the reduced virulence exhibited by *c14dm⁻* mutants, it was not clear whether other cellular processes were also affected. In this study, we revealed profound mitochondrial dysfunctions and elevated level of reactive oxygen species in *c14dm⁻* mutants. These sterol mutants rely heavily on glycolysis to generate energy and are extremely sensitive to glucose restriction. In addition, the accumulation of oxidants appears to be responsible (at least in part) for the previously observed heat sensitivity in *c14dm⁻* mutants. Thus, genetic or chemical inactivation of C14DM can influence the functions of cellular organelles beyond the plasma membrane. These findings shed light on the mechanism of action for azole compounds and provide new insight into the roles of sterol biosynthesis in *Leishmania* parasites.

## Introduction

Parasitic protozoa of the genus *Leishmania* are the causative agents for a group of diseases in humans and animals known as leishmaniasis [1, 2]. During their life cycle, *Leishmania* parasites alternate between flagellated, extracellular promastigotes in sandfly vectors and non-flagellated, intracellular amastigotes in vertebrates. Current drugs for leishmaniasis are plagued with high toxicity, low efficacy and resistance is on the rise [3]. Therefore, research involving the identification and characterization of metabolic pathways that could provide new drug targets is of considerable value.

While mammalian cells synthesize cholesterol as the main sterol, *Leishmania* promastigotes primarily make ergostane-based sterols such as ergosterol and 5-dehydroepisterol [4–6]. Ergosterol is distinguished from cholesterol by the presence of an unsaturated side chain with a methyl group at the C-24 position and an additional double bond at C-7 to C-8 in the B ring of the sterol core. Differing from promastigotes, *Leishmania* amastigotes (which reside in macrophages) mainly salvage cholesterol from the host, while downregulating the de novo ergosterol synthesis (supplemental Figure S14 and S15 in [7]). Despite the low abundance of ergostane-based sterols in amastigotes, sterol synthesis inhibitors and amphotericin B have shown promise as anti-leishmaniasis drugs [5, 8–10], suggesting that the residual amount of endogenous sterols are crucial for amastigotes and/or for the binding of amphotericin B. Among these sterol synthesis inhibitors are azole compounds which primarily target the cytochrome P450 enzyme sterol 14-α-demethylase (C14DM) [5, 11, 12]. Contribution of sterol biosynthesis to *Leishmania* biology is not fully understood, which hinders the development and improvement of azoles as anti-leishmaniasis drugs.

C14DM was found to be refractory to genetic deletion in *Leishmania donovani*, the causative agent for visceral leishmaniasis [13]. In contrast, our earlier study showed that this enzyme was dispensable in *Leishmania major* (responsible for cutaneous leishmaniasis in the old world) [7]. Thus, different *Leishmania* species may possess distinct requirements for sterol synthesis. Importantly, although *L. major* C14DM-null mutants (*c14dm⁻*) are viable as promastigotes under normal tissue culture conditions, they show increased plasma membrane fluidity, disruption in lipid raft formation, and deficiency in lipophosphoglycan (LPG, a major surface glycoconjugate and virulence factor [14]) synthesis [7]. These plasma membrane defects probably contribute to the mutants' hypersensitivity to heat and their severely attenuated virulence in mice, although other factors may be involved as well [7]. A later study on the

sterol C-24-methyltransferase indicate that the accumulation of aberrant, C-14-methylated sterol intermediates, not the lack of ergostane-based sterols, is mainly responsible for the fitness loss displayed by *c14dm⁻* mutants [15].

While sterols are well recognized for their stabilizing effects on the plasma membrane [16, 17], their functions in intracellular organelles are poorly understood, as the end products of sterol synthesis are mainly found on the plasma membrane and secretory vesicles in mammalian cells and yeast [18–20]. In *L. major*, C14DM is primarily found in the ER but also present in the mitochondria suggesting that this enzyme is needed there [7]. An earlier study reported the presence of substantial amounts of endogenous and exogenous sterols in the mitochondria of *Trypanosoma cruzi*, a related trypanosomatid protozoan [21]. Application of sterol biosynthesis inhibitors such as ketoconazole and lovastatin altered the ultrastructure of mitochondria in *T. cruzi*, leading to intense proliferation of the inner-mitochondrial membranes and necrotic cell death [22]. Similarly, RNAi knockdown of ergosterol pathway genes in *Trypanosoma brucei* resulted in disorganization of mitochondria structure and depletion of ATP [23]. In *Leishmania*, the lipid composition of mitochondrial membrane has yet to be determined. Nonetheless, structural and functional abnormalities were also observed among other defects following treatment with azole compounds. For example, in *Leishmania amazonensis*, combinatorial treatment of ketoconazole and terbinafine led to alterations in the distribution and appearance of cristae and the formation of paracrystalline arrays within the mitochondrial matrix [24]. Another study reported the collapse of mitochondrial membrane potential (ΔΨm) following posaconazole treatment in *L. amazonensis* [25]. Besides azoles, treatment of *L. amazonensis* and *T. cruzi* with various azasterols, which inhibit the sterol C-24-methyltransferase, induced a loss of mitochondrial matrix contents [22, 26, 27]. Although the off-target effects from inhibitors cannot be excluded, these observations suggest that sterol synthesis plays important roles in preserving the structure and function of mitochondria in trypanosomatids.

Compared to mammalian cells which usually possess multiple mitochondria per cell, trypanosomatids have only one, relatively large mitochondrion per cell [28]. In addition to energy production, mitochondria in trypanosomatids have been implicated in fatty acid synthesis and redox homeostasis during different stages of their life cycle [29, 30]. Mitochondrial malfunctions, e.g. changes in ΔΨm, disruption of mitochondrial membrane integrity, and increase in reactive oxygen species (ROS) production, are major mediators of cell death under various stress conditions [31–33].

In this study, we sought to address the following important questions. First, how does C14DM inactivation affect the integrity and functions of mitochondria in *L. major*? Second, besides heat sensitivity, do *c14dm⁻* mutants exhibit any other stress response defects? Third, how do mitochondrial deficiencies influence the fitness of *c14dm⁻* parasites? Our results showed that genetic or chemical disruption of C14DM severely compromised the mitochondria in *L. major*, leading to significant ROS accumulation and extreme vulnerability to biologically relevant stress conditions such as heat and glucose starvation. These findings may guide the development of new treatment strategies that exploit the fitness cost associated with drug resistance.

## Results

### *C14dm⁻* promastigotes show increased mitochondrial membrane potential (ΔΨm) and altered mitochondrial morphology

To examine if C14DM deletion affects the mitochondria in *L. major*, we labeled promastigotes with MitoTracker CMXRos, a fluorogenic dye that binds to mitochondria in a ΔΨm-dependent

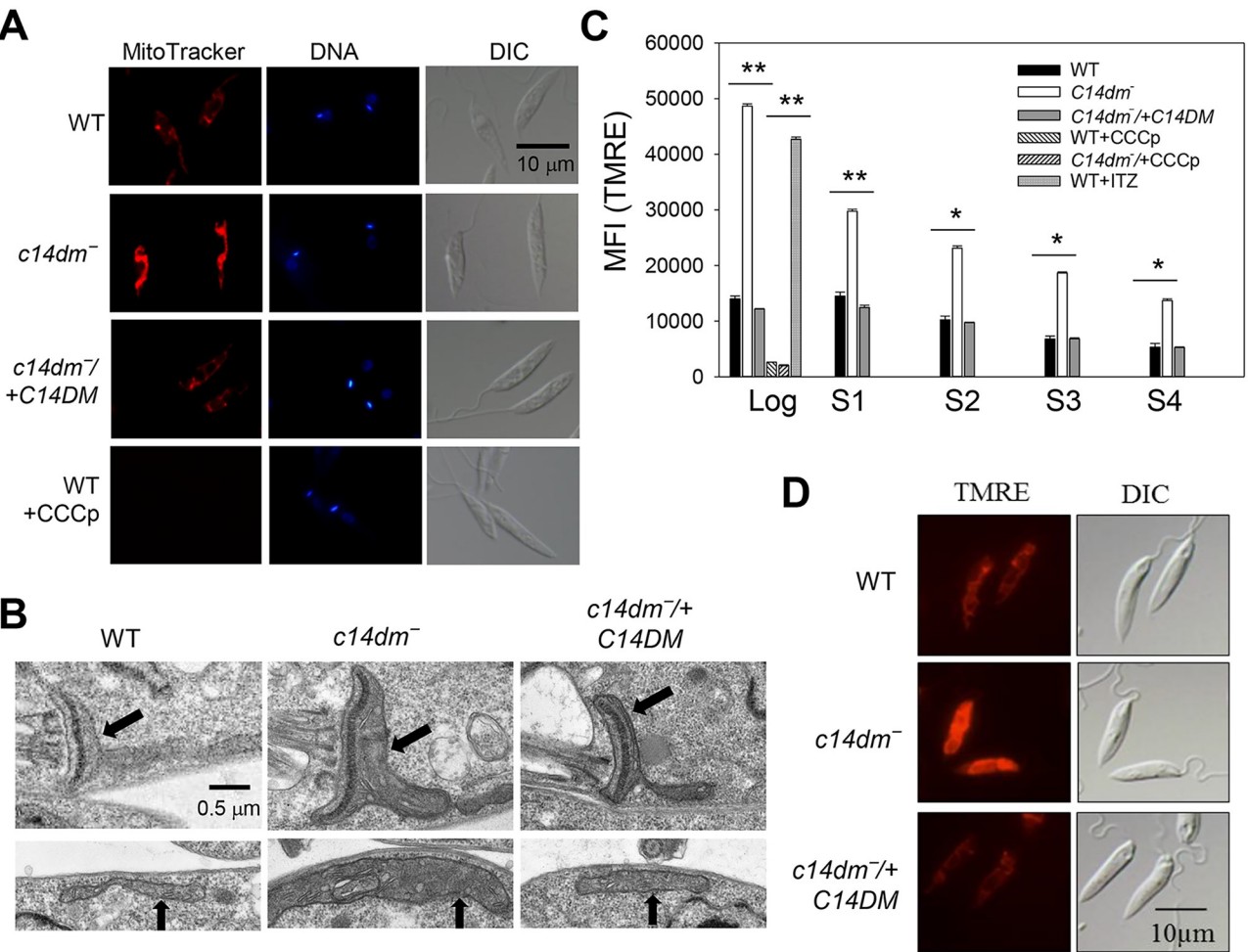

**Fig 1. *C14dm⁻* promastigotes show high mitochondrial membrane potential (ΔΨm).** (**A**) Log phase promastigotes were labeled with 100 nM of Mitotracker CMXRos followed by DAPI staining. WT parasites pretreated with CCCp were included as controls. DIC: differential interference contrast. (**B**) Representative transmission electron micrographs showing mitochondria in log phase promastigotes. Black arrows indicate areas containing kinetoplast (top panel) and tubular mitochondria (bottom panel). (**C**) To measure ΔΨm, log phase and stationary phase (day 1-day 4) promastigotes were labelled with 100 nM of TMRE for 15 min in PBS and mean fluorescence intensity (MFI) were determined by flow cytometry. Controls include WT and *c14dm⁻* parasites pretreated with CCCp and WT cells pretreated with 0.2 μM of ITZ. Bar values represent averages from three experiments and error bars represent standard deviations. (**D**) Live parasites imaging after TMRE labeling.

manner. As indicated in Fig 1A, *c14dm⁻* mutants showed an intense mitochondrial staining 2–4 times stronger than wild type (WT) and *c14dm⁻*/+C14DM (add-back) parasites. In addition, the mitochondria of *c14dm⁻* appeared to be more swollen with larger volume and surface area based on transmission electron microscopy and quantitative analysis by fluorescence microscopy (Fig 1B and S1A Fig). To examine whether the mitochondrial membrane in *c14dm⁻* was compromised, we analyzed the presence of cytochrome *c* in mitochondrial fraction and cytosolic fraction following extraction with digitonin, with ISCL and HSP83 being used as mitochondrial and cytosolic markers, respectively [34] [35] (S1B Fig). Similar to WT and *c14dm⁻*/+C14DM parasites, cytochrome *c* in *c14dm⁻* remained in the mitochondrial enriched fractions when the digitonin extraction was performed at room temperature (S1B Fig). As a control, digitonin treatment at 37 ˚C disrupted the outer mitochondrial membrane, causing cytochrome *c* release into the cytosol [36] (S1B Fig). Thus, despite of its effects on ΔΨm and mitochondrial size,

genetic ablation of C14DM does not affect mitochondrial membrane integrity at least with regard to cytochrome *c* localization under ambient conditions.

Consistent with the MitoTracker staining results, when live parasites were labeled with another ΔΨm-dependent dye tetramethylrhodamine ethyl ester (TMRE), *c14dm̄* mutants retained 2–4 times more fluorescence than WT and *c14dm̄/+C14DM* cells as indicated by flow cytometry and microscopy (Fig 1C and 1D). The difference between WT and *c14dm̄* parasites was more pronounced during the log phase when cells were metabolically active than stationary phase (Fig 1C). Similar results were observed after staining with rhodamine 123 (S2A Fig). As expected, pretreatment with the mitochondrial uncoupler CCCp (75 μM for 15 min) dramatically reduced the incorporation of TMRE and MitoTracker into WT and *c14dm̄* parasites (Fig 1A and 1C), without significant effect on cell viability (<5% of cells were propidium iodide positive by flow cytometry). Furthermore, WT promastigotes grown in the presence of itraconazole (ITZ, a C14DM inhibitor) [7] for 48 hours at sublethal concentrations showed similar ΔΨm as *c14dm̄* mutants (Fig 1C). Finally, we also examined if extracellular nutrient level affects ΔΨm by labelling promastigotes with TMRE in either complete M199 medium, M199 medium without fetal bovine serum, phosphate-buffered saline (PBS) supplemented with 5.5 mM of glucose (the glucose concentration in M199 medium), or PBS alone (Fig 2A). In these experiments, *c14dm̄* mutants exhibited 2.5–4 times higher ΔΨm than WT/add-back parasites and no significant differences were detected between nutrient-replete and nutrient limiting conditions (Fig 2A). Collectively, these observations suggest that C14DM inhibition leads to significantly increased ΔΨm in *L. major* promastigotes regardless of extracellular nutrient levels.

## Inactivation of C14DM leads to ΔΨm-dependent accumulation of ROS in the mitochondria

In mammalian cells, a high ΔΨm may cause accumulation of ROS in the mitochondria (mainly in the form of superoxide or $O_2^{\bullet-}$) in part due to reverse electron flow from complex II to complex I of the electron transport chain (ETC) [37–39]. To assess the ROS level in *c14dm̄* mutants, promastigotes were labelled with a mitochondria-specific ROS indicator MitoSox Red which binds to mitochondrial nucleic acid and fluoresces upon oxidation by superoxide [40]. In complete M199, *c14dm̄* mutants showed 3.5 times more MitoSox labeling than WT and add-back cells (Fig 2B). Unlike ΔΨm, the mitochondrial ROS level in *c14dm̄* was sensitive to nutrient availability (Fig 2B): PBS only (the highest) > PBS with glucose > M199 medium without fetal bovine serum > complete M199 (the lowest). Importantly, pretreatment of *c14dm̄* parasites with CCCp significantly reduced MitoSox Red labeling, indicating that the high ΔΨm is mainly responsible for ROS accumulation (Figs 2A, 2B and 1C). These findings suggest that the mitochondria in *c14dm̄* mutants are under significant oxidative stress during starvation.

Next we examined the mitochondrial ROS levels in log phase and stationary phase promastigotes after labeling them with MitoSox in PBS alone (Fig 2C). Compared to WT and add-back cells, *c14dm̄* mutants contained 10–15 times more mitochondrial ROS by flow cytometry during log phase and early stationary phase, and the difference became less dramatic during late stationary phase when cells ceased proliferation (Fig 2C). Microscopic examination of MitoSox-labelled parasites revealed singular fluorescent spots in every cell representing the kinetoplast DNA, indicating the mitochondrial origin of ROS (Fig 2D). In these images, MitoSox signals from *c14dm̄* were much more prominent than those from WT and *c14dm̄/+C14DM* cells (Fig 2D), which was in agreement with flow cytometry analysis (Fig 2B and 2C). Furthermore, when cultivated in the presence of ITZ (at 0.1–0.2 μM or near EC25 levels), *L.*

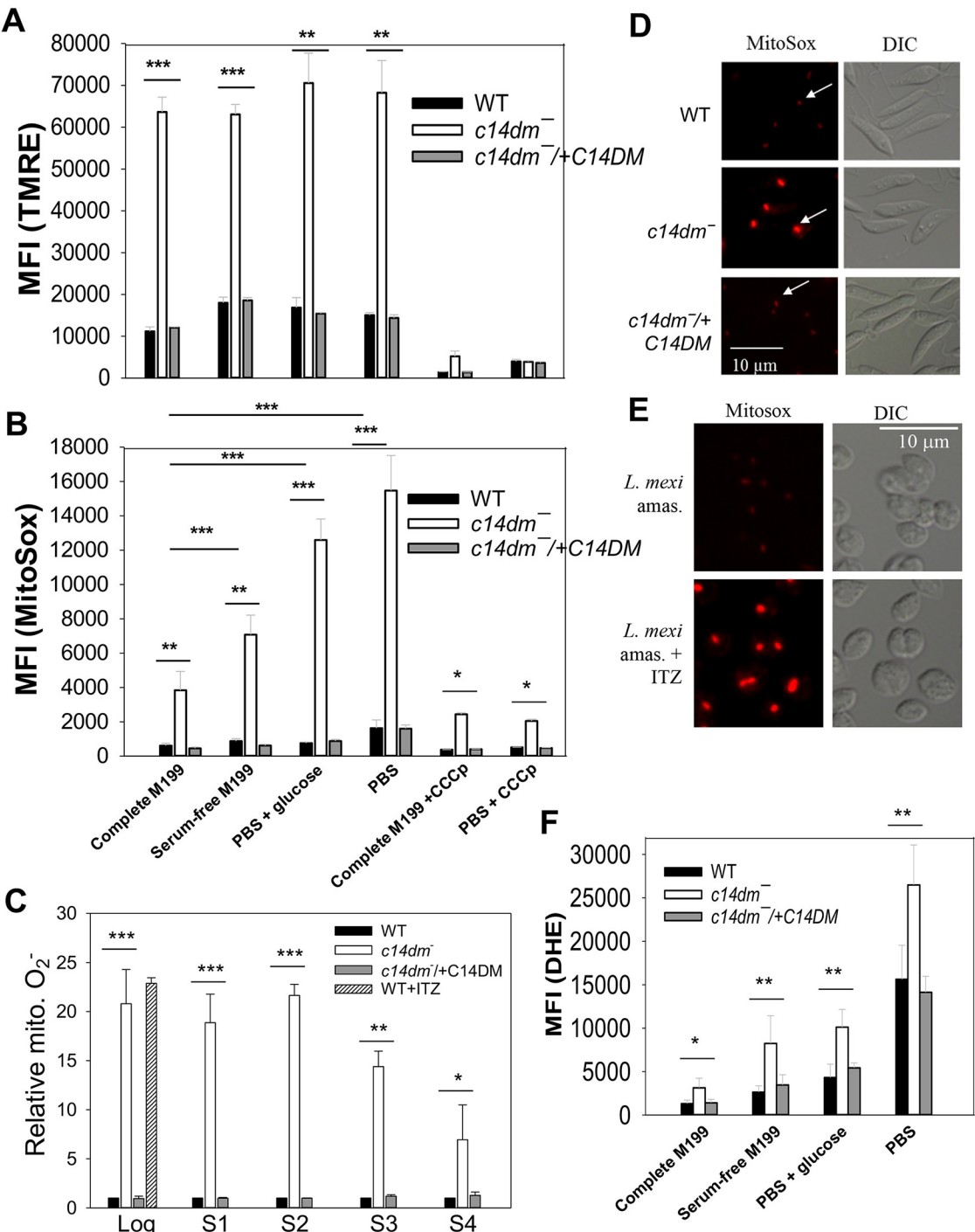

**Fig 2. Inactivation of C14DM leads to ROS accumulation in the mitochondria.** (**A**, **B** and **F**) Promastigotes were incubated in complete M199, serum-free M199, PBS contain 5.5 mM of glucose, or PBS only and labeled with 100 nM of TMRE (**A**, for 15 min), 5 μM of MitoSox Red (**B**, for 25 min) or 5 μM of DHE (**F**, for 30 min). MFIs were determined by flow cytometry. Effects of CCCp (75 μM, 15 min) were also monitored in **A** and **B**. (**C**) Log or stationary phase (day 1-day 4) promastigotes were labeled with 5 μM of MitoSox Red for 25 min in PBS and MFIs were determined by flow cytometry and normalized to values from log phase WT. WT cells grown in the presence of ITZ (for 48 hours) were also analyzed. (**D** and **E**) Live cell imaging of *L. major* promastigotes (**D**) or *L. mexicana* axenic amastigotes (grown in the absence or presence of 0.1 μM of ITZ for 48 hours, **E**) after labelling with MitoSox Red in PBS. White arrows indicate kDNA binding of MitoSox. (**E**) Live after labeling with MitoSox Red. In **A-C** and **F**, averaged values from five experiments are shown and error bars represent standard deviations.

*major* WT promastigotes and *L. mexicana* axenic amastigotes displayed high levels of MitoSox Red staining similar to *c14dm⁻* (Fig 2C and 2E), indicating that C14DM inhibition causes ROS built-up in the mitochondria of *Leishmania* parasites. In comparison to *c14dm⁻*, mitochondrial defects exhibited by the sterol C-24-methyltransferase mutants (*smt⁻*) were less profound (S2 Fig) [15], suggesting that the accumulation of C-14-methylated sterols is mainly responsible for these changes.

We then determined the cytoplasmic ROS level in log phase promastigotes using dihydroethidium (DHE), a cytosolic ROS indicator. As illustrated in Fig 2F, parasites showed high DHE staining under nutrient-limiting conditions. In addition, the amount of cytosolic ROS in *c14dm⁻* was about two-fold higher than WT and add-back parasites, which was not as dramatic as the increase of mitochondrial ROS (Fig 2B and 2F). As a control, treatment of WT parasites with hydrogen peroxide (100 µM for 10 min) led to significantly increased staining for DHE but not MitoSox Red, confirming the specificity of MitoSox Red for mitochondrial superoxide (S3 Fig). Together, these data suggest that genetic or chemical ablation of C14DM leads to significant accumulation of ROS inside the mitochondria.

## Examination of superoxide dismutase (SOD) expression in *c14dm⁻* mutants

As byproducts of mitochondrial respiration, ROS (mainly the form of $O_2^{\bullet-}$) can be generated from the partial reduction of oxygen by complexes I, II, and III of the ETC [41]. To limit the damaging effects of ROS, *Leishmania* cells utilize several antioxidant systems involving superoxide dismutases (SODs) and tryparedoxin peroxidases [42, 43]. The fact that *c14dm⁻* parasites accumulate high levels of $O_2^{\bullet-}$ suggests that either the ROS removal system is defective, or the production of $O_2^{\bullet-}$ is elevated due to excessive electron leakage from the ETC in these mutants. Here we examined whether a dysregulation of SOD, the primary enzyme to detoxify $O_2^{\bullet-}$, is responsible for the accumulation of mitochondrial ROS in *c14dm⁻*. *Leishmania* encode three iron-dependent SODs: the mitochondrial SODA and two glycosomal SODB (SODB1 and SODB2) [43, 44]. In whole cell lysate, we detected a 14–98% increase in SOD activity from *c14dm⁻* mutants and the increase was more pronounced during the stationary phase (Fig 3A). Western blot revealed that *c14dm⁻* mutants had similar levels of SODA (mitochondrial) and SODB (cytosolic) protein as WT parasites (Fig 3B). The increase in SOD activity in late stationary phase *c14dm⁻* coincided with the decrease of mitochondrial ROS (although still higher than WT and add-back parasites, Fig 2C, S2B Fig and Fig 3A), suggesting an adaptive response from the mutants to upregulate their antioxidant defense machinery. Collectively, these findings indicate that the ROS is accumulated in *c14dm⁻* mutants despite of near-normal SOD production.

## *C14dm⁻* mutants show reduced cellular respiration and increased glycolytic ATP production

The mitochondrial abnormalities in *c14dm⁻* prompted us to examine if these mutants could effectively carry on respiration and generate ATP. First, we measured the cellular oxygen consumption rate using the MitoXpress oxygen probe as described previously [45]. When incubated a mitochondrial respiration buffer made of Hanks' balanced salt solution (HBSS) supplemented with sodium pyruvate and 2-deoxy-D-glucose (to block glycolysis) [46], *c14dm⁻* parasites showed a significant reduction (42–60%) in oxygen consumption compared to WT cells, and such defect was corrected in the *c14dm⁻*/+C14DM parasites (Fig 4A and 4B). As expected, the addition of antimycin A, a respiratory inhibitor that blocks cytochrome *c* reductase or complex III [47], decreased oxygen consumption in WT parasites (Fig 4A). A similar experiment was performed when cells were incubated in the Dulbecco's Modified Eagle's

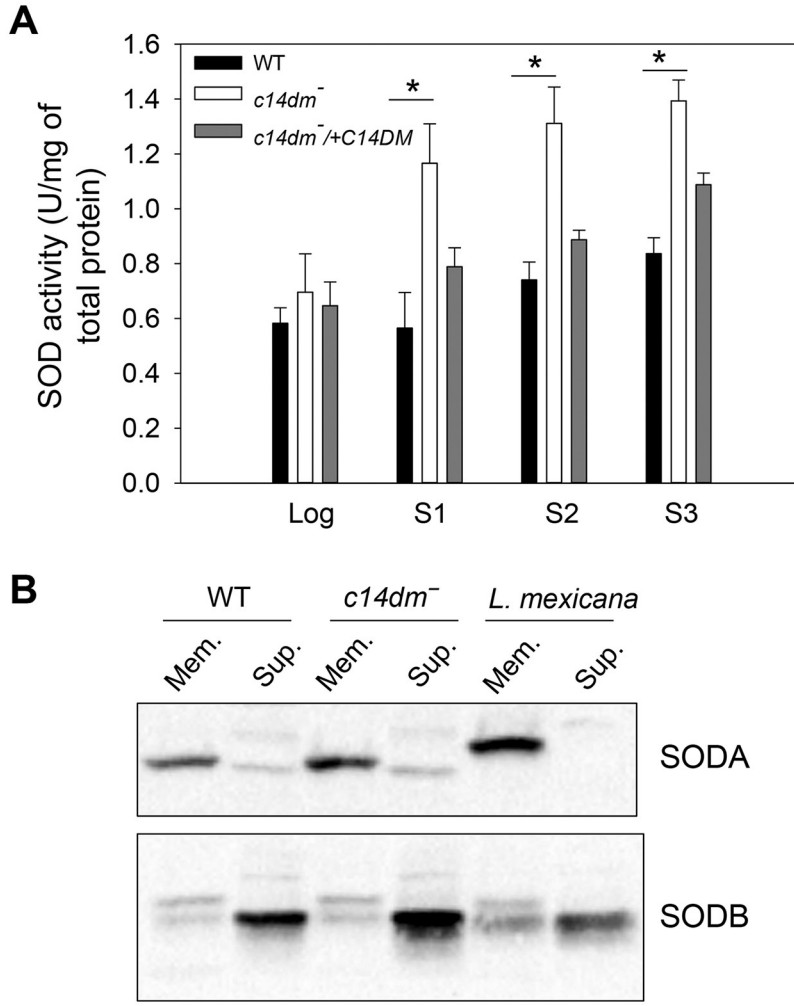

**Fig 3. Increased superoxide dismutase (SOD) activity in *c14dm⁻* parasites.** (**A**) Whole cell lysates from log phase and stationary phase (day 1-day 3) promastigotes were collected as described in **Materials and Methods**. SOD activity was measured using a SOD assay kit and results were normalized based on the amount of protein in each sample. Error bars represent standard deviations from two independent experiments. (**B**) Mitochondria enriched membrane fractions (Mem.) and cytosolic fractions (Sup.) were isolated from log phase promastigotes as described. Western blots were performed using antibodies against *L. amazonensis* SODA (mitochondrial) and SODB (both SODB1 and SODB2, glycosomal).

Medium (DMEM) which contained 25 mM of glucose. While all cells exhibited higher oxygen consumption rates in DMEM than in HBSS, *c14dm⁻* mutants again used less oxygen than WT and *c14dm⁻*/+C14DM parasites (Fig 4C). Given the considerable ROS accumulation in *c14dm⁻*, the low oxygen consumption rate may protect these mutants from further oxidative damage.

Next, we determined the ATP production from WT, *c14dm⁻* and *c14dm⁻*/+C14DM parasites. When log phase promastigotes were incubated in an isotonic buffer (without glucose or any inhibitor) for 1 hour, a 30–40% increase in ATP content was observed in *c14dm⁻* mutants in comparison to WT or add-back parasites (Fig 4D, lane 1). Supplementation with glucose led to increased ATP production in all cells, presumably from both glycolysis and mitochondrial respiration (Fig 4D, lane 4). Again, despite the low oxygen consumption, *c14dm⁻* mutants produced 30–40% more ATP than WT and add-back parasites from glucose (Fig 4D, lane 4), suggesting that these cells have increased capacity to generate energy through glycolysis.

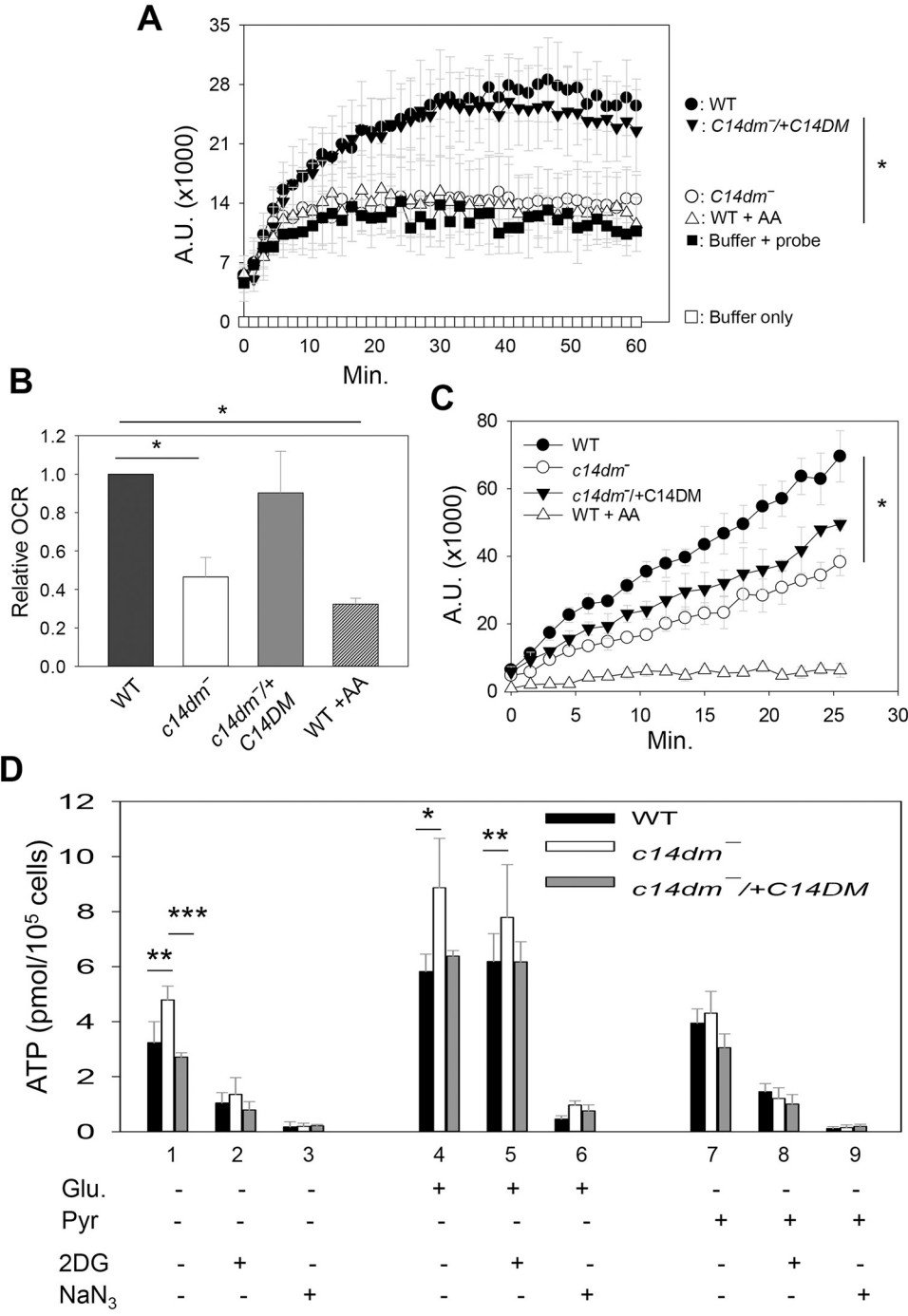

**Fig 4. *C14dm⁻* parasites show compromised cellular respiration.** (**A**-**C**) Log phase promastigotes were incubated in a mitochondrial respiration buffer (**A**-**B**) or DMEM (**C**) and oxygen consumption rates (OCR) were determined using the MitoXpress probe as described in **Materials and Methods**. Buffer only and buffer with probe (no cells) were included as blanks (**A**). In **A** and **C**, fluorescence signals (in arbitrary units or A.U.) were measured every 90 s. In **B**, the relative OCR from the experiments in **A** was calculated by subtracting the 0 min reading from the 15 min reading. WT parasites treated with antimycin A (AA) were included as controls. (**D**) Cellular ATP contents in log phase promastigotes were measured after 1 hour incubation in an assay buffer (21 mM of HEPES, 0.7 mM of Na$_2$HPO4, and 137 mM of NaCl, pH 7.2) in the absence or presence of glucose (Glu, 5.5 mM), sodium pyruvate (Pyr, 5.5 mM), 2-deoxy-D-glucose (2DG, 5.5 mM), or sodium azide (NaN$_3$, 20 mM). All experiments were repeated three or four times and error bars represent standard deviations.

Curiously, addition of sodium pyruvate had only modest effect on ATP production (Fig 4D, lane 7). Although *Leishmania* parasites possess putative plasma membrane-bound pyruvate transporters [48], the efficiency of pyruvate uptake is not known. Another possibility is that parasites can sense the lack of carbon sources besides pyruvate and regulate ATP biogenesis accordingly. As expected, addition of 2-deoxy-D-glucose (5.5 mM) significantly reduced ATP production in the absence of exogenous glucose (Fig 4D, lane 2 and 8). No significant reduction was detected when cells were incubated in the presence of glucose (5.5 mM), which might be due to inadequate competition on glucose uptake and utilization (Fig 4D, lane 5). Furthermore, addition of sodium azide which inhibits cytochrome *c* oxidase [46, 49] dramatically lowered ATP production in all cells (Fig 4D, lane 3, 6 and 9), suggesting that the glycolysis flux may be coupled to the efficiency of oxidative phosphorylation respiration in *Leishmania* so that cells can turn down glycolysis to prevent the buildup of toxic metabolites.

### Glucose starvation has deleterious effect on *c14dm⁻* parasites

Our findings so far suggest that in comparison to WT cells, *c14dm⁻* mutants depend more on non-respiration pathways such as glycolysis to generate energy. If so, these mutants would be highly sensitive to glucose restriction. To test this possibility, we exposed parasites to glucose-limiting conditions and measured cell survival over time. As illustrated in Fig 5A, when incubated in HBSS, *c14dm⁻* mutants showed substantial cell death after 12 hours. At 20–24 hours, more than 90% of *c14dm⁻* cells were dead (Fig 5A and 5B). Similar results were observed when parasites were incubated in other neutral, isotonic buffers such as PBS (S4 Fig). In addition, *L. major* WT parasites cultivated in the presence of ITZ were highly vulnerable to glucose starvation like *c14dm⁻* mutants (S5 Fig). In these experiments, WT control, *smt⁻*, *smt⁻*/+SMT and *c14dm⁻*/+C14DM parasites showed very good survival (<9% death after 24 hours), indicating this phenotype is caused by the ablation of C14DM (Fig 5B). Importantly, supplementation with glucose at 5.5 mM (the glucose concentration in M199 medium) completely rescued *c14dm⁻* mutants, whereas at the same molar concentration, sodium pyruvate (a respiration substrate), 2-deoxy-D-glucose (a non-metabolizable glucose analog), or aspartate (a gluconeogenic amino acid) failed to protect the mutants (Fig 5). Further tests revealed that glucose concentrations above 100 μM were sufficient to rescue *c14dm⁻* mutants (S4 Fig).

Recent studies have revealed that bloodstream forms of *Trypanosoma brucei* can use glycerol to support growth in the absence of glucose [50, 51]. To test if *Leishmania* parasites can utilize glycerol effectively, we attempted to rescue *c14dm⁻* mutants with glycerol. Remarkably, when provided at more than 100 μM, glycerol was able to protect *c14dm⁻* mutants under glucose-free conditions (Fig 5 and S4 Fig). Both glucose and glycerol were also able to alleviate the mitochondrial defects in *c14dm⁻* mutants (S6 Fig). Together, our results indicate that inactivation of C14DM leads to hypersensitivity to glucose starvation, which is likely caused by energy depletion and excessive ROS accumulation.

### *C14dm⁻* parasites show increased uptake of glucose and glycerol

Next, we examined whether the uptake of glucose and glycerol is altered in *c14dm⁻* parasites using radiolabeled substrates as previously described [52, 53]. If respiration defects force *c14dm⁻* mutants to rely more on glycolysis to generate energy, these parasites may have higher demands for glucose or equivalent carbon sources. Trypanosomatid protozoans including *Leishmania* species possess a family of glucose transporters [54] and aquaglyceroporins [53] that can mediate nutrient uptake. As shown in Fig 6, *c14dm⁻* mutants were more efficient than

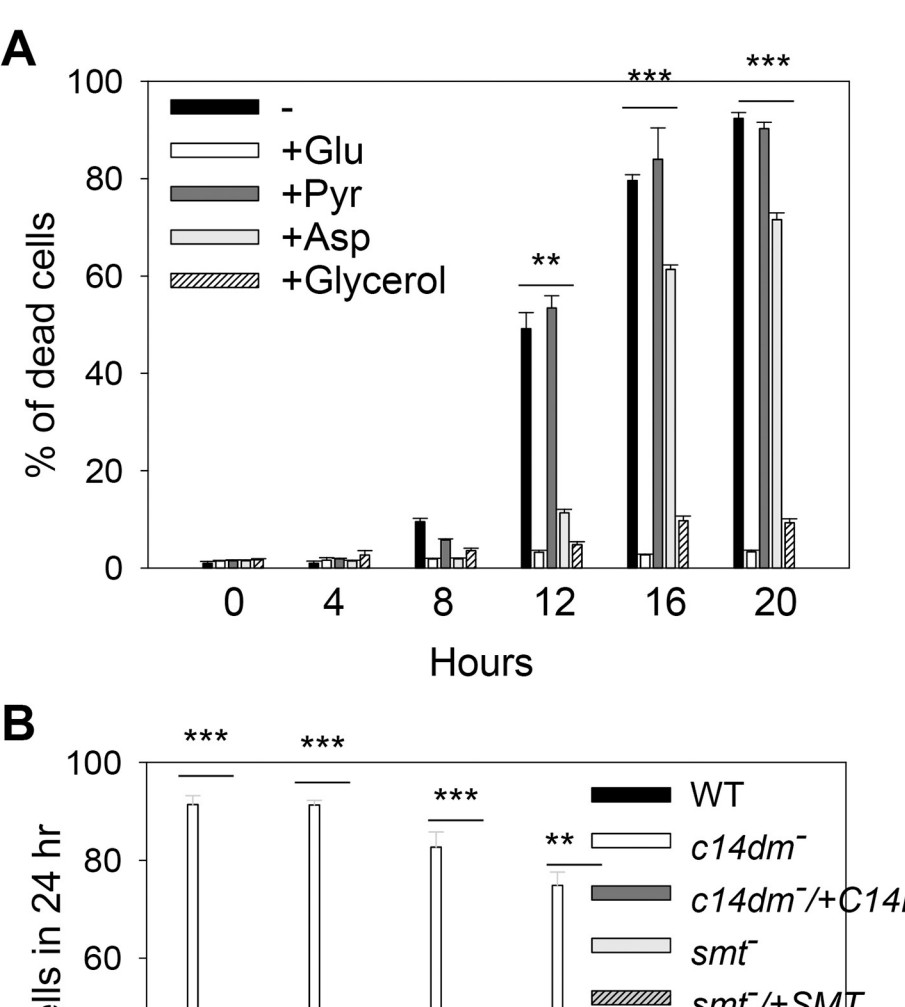

**Fig 5. *C14dm⁻* parasites are hypersensitive to starvation.** Log phase promastigotes were inoculated in HBSS in the absence or presence of 2DG (5.5 mM), Pyr (5.5 mM), aspartate (Asp, 5.5 mM), Glu (5.5 mM), or glycerol (0.1 mM). In **A**, percentages of cell death for *c14dm⁻* mutants were determined at the indicated time points. In **B**, percentages of cell death were determined after 24 hours for WT, *c14dm⁻*, *c14dm⁻/+C14DM*, *smt⁻*, and *smt⁻/+SMT* parasites. Error bars represent standard deviations from three independent experiments.

WT and add-back parasites in the uptake of glucose and glycerol. The incorporation of glucose was much more robust at room temperature than at 4 ˚C (Fig 6A), whereas the incorporation of glycerol was less dependent on temperature (Fig 6B and 6C). The increased uptake of glucose and glycerol in *c14dm⁻* may serve as a compensatory mechanism to offset deficiencies in mitochondrial respiration.

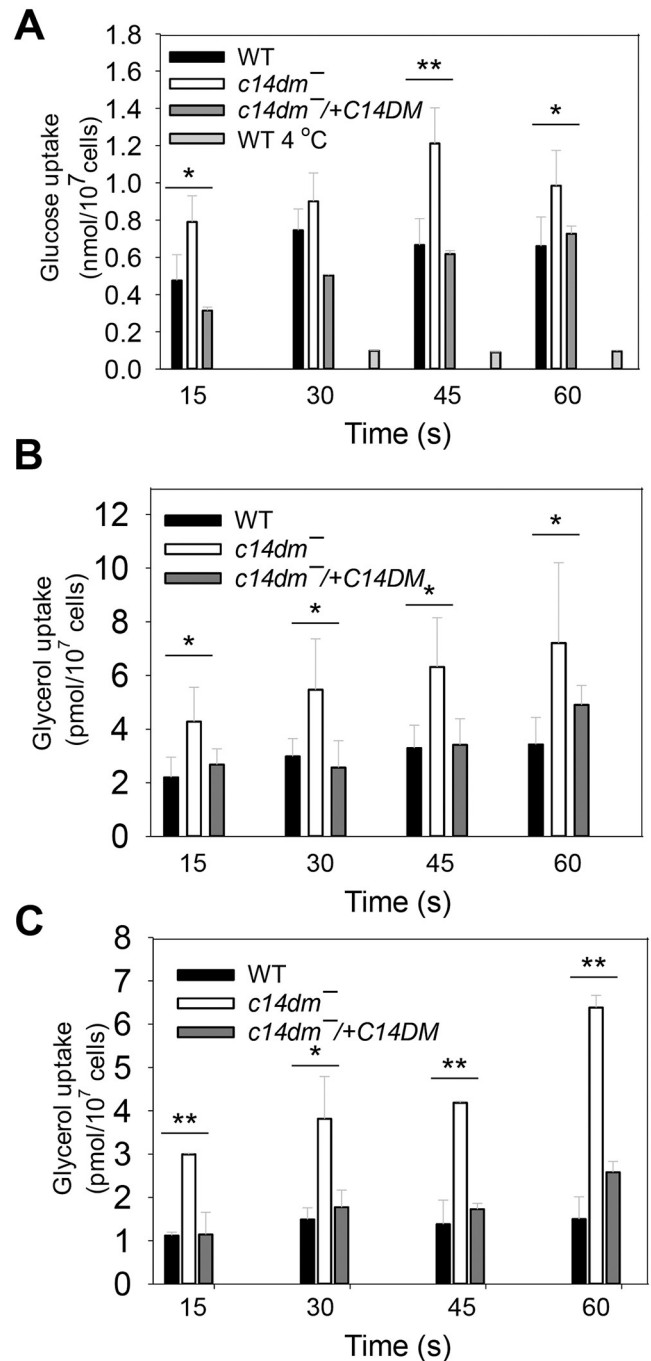

**Fig 6. *C14dm⁻* parasites show increased uptake of glucose and glycerol.** Incorporation of radiolabeled glucose (**A**) or glycerol (**B** and **C**) were determined as described under **Materials and Methods**. Transport assays were performed at room temperature (**A** and **B** unless specified) or 4 ˚C (**C**). Error bars represent standard deviations from three independent experiments.

## ROS accumulation is partially responsible for the heat sensitivity in *c14dm⁻* mutants

We have previously shown that *c14dm⁻* parasites are extremely sensitive to heat stress (e.g. 37 ˚C treatment), and their increased plasma membrane fluidity (induced by the accumulation of

C-14-methylated sterol intermediates) is a likely cause [7]. To investigate if the accumulation of ROS is also a contributing factor, WT and *c14dm⁻* parasites were incubated in PBS at either 27 ˚C (the regular temperature for promastigotes) or 37 ˚C. After 3 hours, a marked reduction in ΔΨm was detected in 30–35% of *c14dm⁻* parasites without substantial cell death (Fig 7A and 7B). No significant change in ΔΨm was observed in WT parasites (Fig 7B). To determine if the loss of ΔΨm was associated with compromised mitochondrial membrane integrity, we assessed the release of cytochrome *c* from mitochondria into the cytosol (Fig 7C and 7D). For WT cells, the majority of cytochrome *c* was found in the mitochondrial enriched membrane fractions under control (27 ˚C) and heat stress (37 ˚C, 3 hours) conditions (Fig 7C and 7D). In contrast, ~40% of cytochrome *c* was detected in the cytosolic fraction when *c14dm⁻* parasites were subjected to heat treatment (Fig 7C and 7D). Since 3 hours incubation at 37 ˚C did not induce substantial cell death (Fig 7A), this finding indicates that the mitochondrial disruption occurred when the majority of *c14dm⁻* mutants were viable. In addition, we measured the levels of cytosolic ROS in control and heat stressed cells. Unlike WT and add-back parasites which showed no significant change, *c14dm⁻* mutants displayed ~3-fold increase in cytosolic ROS following 37 ˚C treatment for 3 hours (Fig 7E). Together, these data suggest that heat stress causes a partial rupture of mitochondrial outer membrane in *c14dm⁻*, leading to the dissemination of cytochrome *c* and ROS into the cytosol.

To further evaluate the contribution of ROS accumulation to the observed heat response phenotype, we treated *c14dm⁻* mutants with glutathione (GSH, an antioxidant) before exposing them to 37 ˚C. When administered at 5 mM, GSH treatment of *c14dm⁻* reduced the mitochondrial ROS by ~60% and the heat-induced cell death in 8 hours by ~50% (Fig 7F and 7G). The effect was much less prominent when GSH was provided at 1 mM (Fig 7F and 7G), indicating a dosage-dependent alleviation of ROS stress. Additionally, we tested the effect of antimycin A, an oxidative stress inducer (S3A Fig), on the heat sensitivity of WT parasites. At 27 ˚C, 10 μM of antimycin A had very mild effect on cell survival (<10% death after 36 hours) but at 37 ˚C, more than 60% of WT cells died after 8 hours (Fig 7H). From these studies, we conclude that in addition to plasma membrane defects, oxidative stress also makes *c14dm⁻* mutants vulnerable to heat.

### *C14dm⁻* mutants exhibit increased sensitivity to pentamidine

Finally, to probe whether the mitochondrial defects in *c14dm⁻* can be exploited for developing better chemotherapy, we tested the susceptibility of these parasites to pentamidine, an antimicrobial agent with efficacy against trypanosomatids [55]. While the mode of action for pentamidine is not completely understood, previous reports suggest that this drug needs to be sequestered into the mitochondria of *Leishmania* in a ΔΨm-dependent manner to exert its activity [55, 56]. Thus, the elevated ΔΨm may make *c14dm⁻* more vulnerable to pentamidine. Indeed, in a 48 hours drug response assay [15], *c14dm⁻* mutants showed heightened growth inhibition by pentamidine, as their EC90, EC50 and EC25 (effective concentrations required to inhibit growth by 90%, 50% and 25% respectively) were 5–15 times lower than those of WT and add-back parasites (Fig 8A). WT cells grown in the presence of ITZ were also hypersensitive to pentamidine, suggesting that these two drugs can work synergistically (Fig 8A). In addition, we examined whether short time exposure to pentamidine can cause death or irreversible damage to *c14dm⁻* mutants by treating parasites with 2.5 μM or 25 μM of pentamidine for one hour, followed by two washes with PBS and then let cells grow in drug-free M199 media [57]. As shown in Fig 8B, exposure to 2.5 μM of pentamidine for one hour led to significant grow delay (2–3 days) in *c14dm⁻* and these mutants could only grow to 5–9 x 10⁶ cells/ml in six days (the maximal density for *L. major* promastigotes in complete M199 is ~3 x 10⁷ cells/ml). In

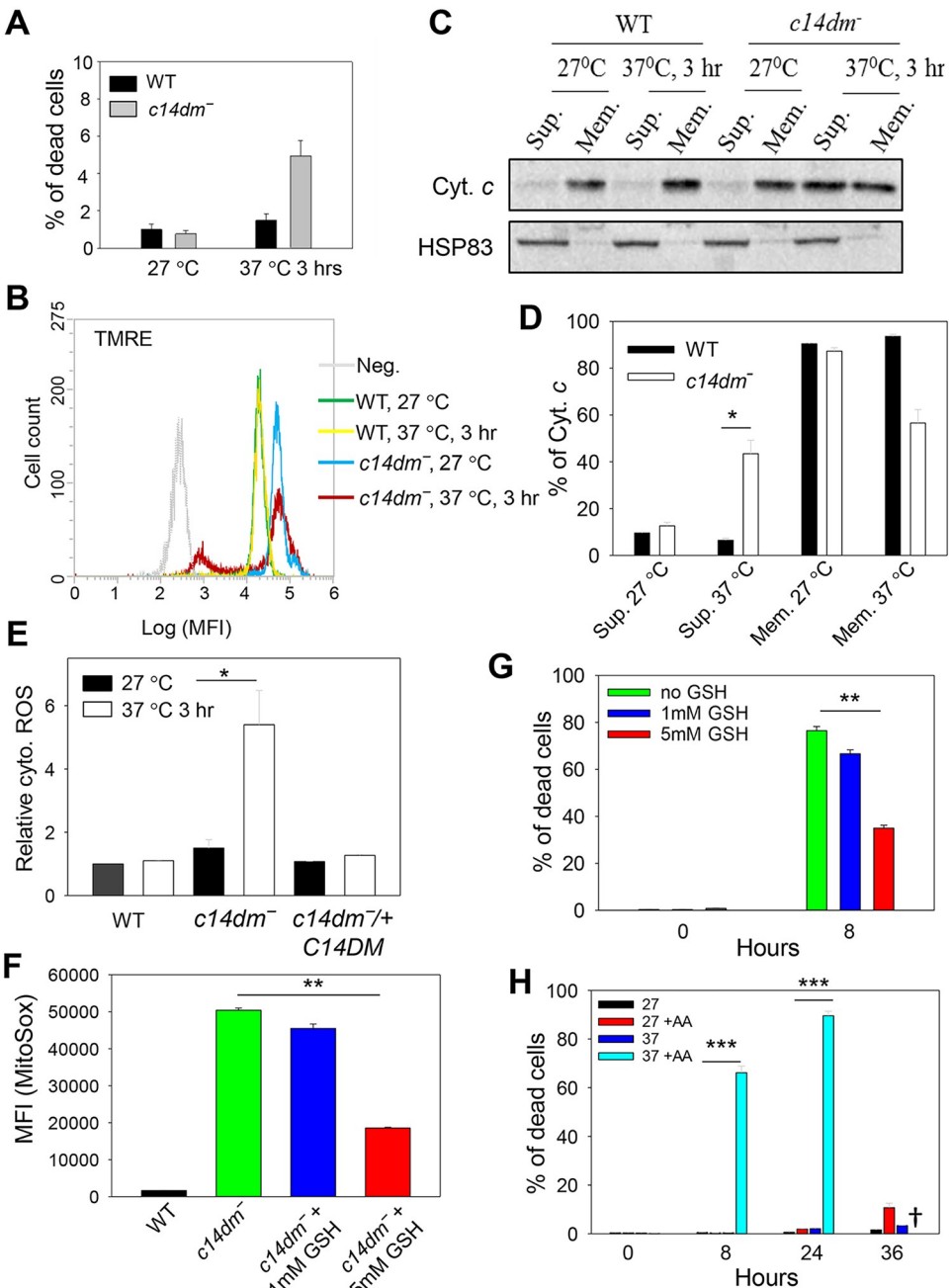

**Fig 7. ROS accumulation contributes to heat sensitivity in *c14dm⁻* mutants.** (**A**-**E**) Log phase promastigotes were incubated in PBS at 27 ˚C or 37 ˚C for 3 hours. In **A**, percentages of cell death were determined by flow cytometry after PI staining. In **B**, ΔΨm was determined after TMRE labeling. Unlabeled WT parasites were included as a negative control (Neg). In **C**-**D**, parasites were lysed and cytosolic fractions (Sup.) were separated from mitochondrial fractions (Mem.) as described in **Materials and Methods**. Distribution of cytochrome *c* and HSP83 (a cytosolic protein marker) were determined by Western blots (**C**) and quantified (**D**). In **E**, parasites were labelled with the cytosolic ROS indicator DHE. Fluorescence intensities were measured by flow cytometry and relative cytosolic ROS levels were determined. (**F**) Log phase promastigotes were incubated at 37 ˚C in the absence or presence of reduced GSH. After 8 hours, mitochondrial ROS levels were measured as described. (**G**) *C14dm⁻* parasites were incubated at 37 ˚C in the absence or presence of reduced GSH and percentages of cell death were determined at 0 and 8 hours. (**H**) WT parasites were incubated at 27 ˚C or 37 ˚C in the absence or presence of AA (10 µM) and percentages of dead cells were determined by flow cytometry at the indicated time points. †: Very few WT+AA cells were detectable after 36 hours at 37 ˚C. Error bars represent standard deviations from three independent experiments.

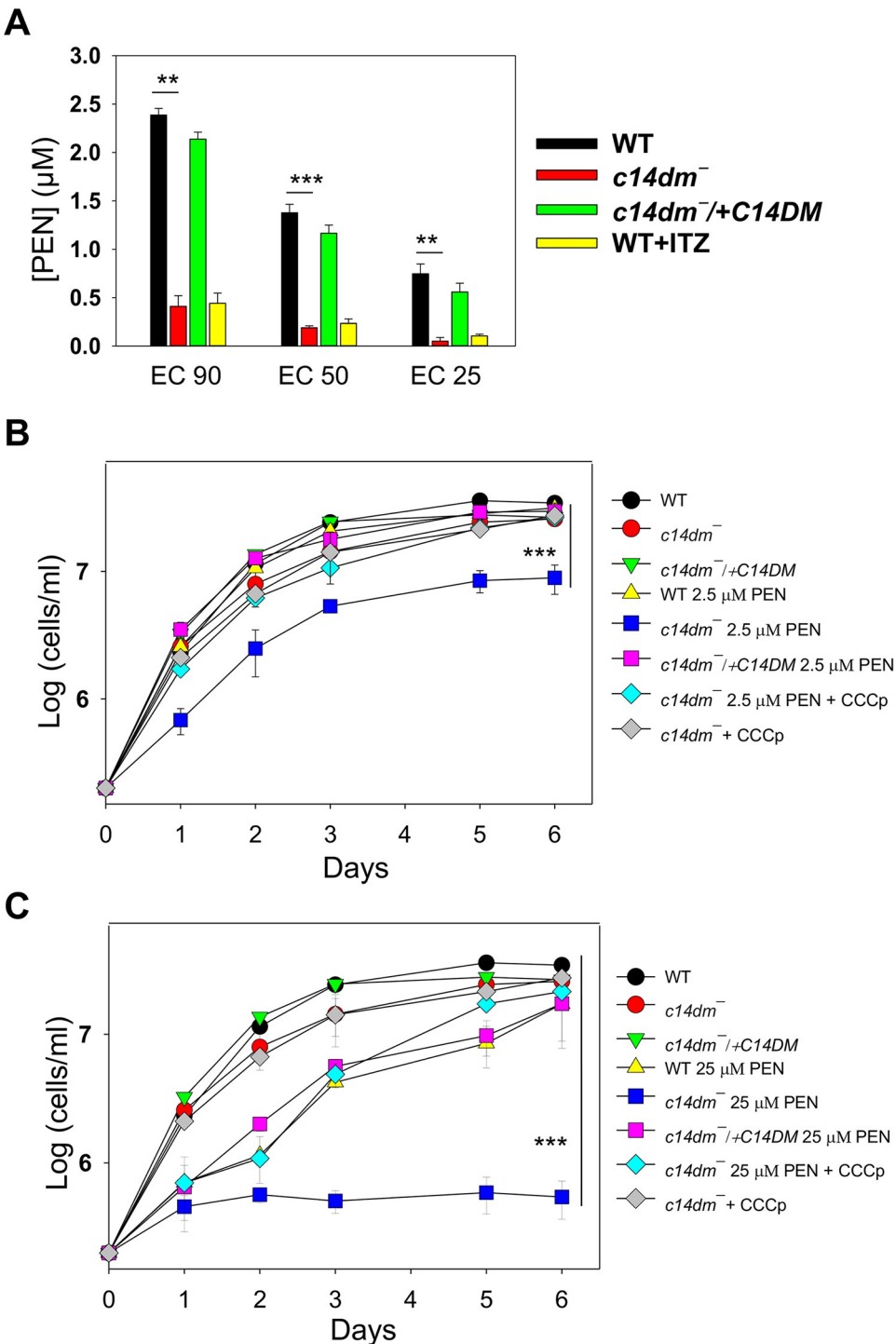

**Fig 8. *C14dm⁻* parasites show increased sensitivity to pentamidine (PEN).** (**A**) Log phase promastigotes were inoculated in M199 media containing various concentrations of PEN. Culture densities were determined after 48 hours and the PEN concentrations required to inhibit growth by 90%, 50%, and 25% were determined (as EC90, EC50 and EC25 respectively) using cells grown in the absence of PEN as controls. "WT + ITZ" represent WT parasites grown in the presence of 0.5 μM of ITZ. (**B-C**) Log phase promastigotes were inoculated in M199 media containing 2.5 μM (**B**) or 25 μM (**C**) of PEN for one hour, followed two washes with PBS. Cells were then re-inoculated in M199 media and culture densities were determined every 24 hours. Some *c14dm⁻* mutants were also incubated in the presence of 75 μM of CCCp along with PEN (for one hour). Parasites originally inoculated in the absence of PEN were included as controls. Error bars represent standard deviations from three or four independent experiments.

contrast, for WT and add-back parasites, 2.5 μM of pentamidine had little impact and it took 25 μM of pentamidine to cause a similar degree of inhibition (Fig 8B and 8C). Meanwhile, 25 μM of pentamidine was sufficient to halt the growth of *c14dm⁻* nearly completely after the one-hour exposure (Fig 8C). CCCp treatment was able to rescue the mutants in this wash out experiment (Fig 8B and 8C), suggesting that the high ΔΨm and/or mitochondrial ROS are responsible the hypersensitivity of *c14dm⁻* to pentamidine.

## Discussion

Our previous work has revealed significant plasma membrane defects in *c14dm⁻* mutants including increased membrane fluidity and failure to maintain detergent resistant membrane fractions [7]. While predominantly located in the plasma membrane, sterols are also present in substantial amounts in lipid droplets and intracellular organelle membranes [20, 58]. A more thorough characterization of the *c14dm⁻* mutants would reveal new insight into the roles of sterol metabolism in *Leishmania* and provide new leads to exploit the vulnerability caused by the accumulation of abnormal sterols. In this study, we found a 2–5 fold increase in the ΔΨm of *c14dm⁻* relative to WT and add-back parasites and similar results were observed in ITZ-treated WT parasites (Fig 1 and S2 Fig). We also detected a modest swelling of mitochondria in *c14dm⁻* mutants (Fig 1B and S1A Fig). These findings are consistent with the previously reported hyperpolarization of mitochondrial membrane in *Leishmania mexicana* following treatment with triazole compounds [59].

In mammalian cells, elevated ΔΨm may cause electron flow in the reverse direction through the ETC leading to electron leakage and superoxide accumulation in the mitochondrial matrix [37, 39]. By flow cytometry and fluorescence microscopy, we observed significantly higher levels of superoxide in the mitochondria of *c14dm⁻* mutants and ITZ-treated WT parasites in comparison to control cells (Fig 2 and S2 and S3 Figs). Under the nutrient-replete condition, a 3-5-fold increase was detected in *c14dm⁻* mutants. Under the nutrient-limiting condition, the difference became 10-12-fold (Fig 2B). Meanwhile, the cytosolic ROS level in *c14dm⁻* mutants was only about 2-fold higher than WT and add-back parasites, suggesting that the majority of accumulated ROS was sequestered in the mitochondria under ambient temperature (Fig 2F and S3 Fig). In agreement with this notion, the mitochondrial membrane integrity in *c14dm⁻* appeared to be intact based on the intracellular distribution of cytochrome *c* (S1B Fig). It is not clear why inactivation of C14DM leads to elevated levels of ΔΨm and mitochondrial ROS. The expression and activity of SODs were not compromised in *c14dm⁻* mutants (Fig 3), suggesting that ROS production was higher in the absence of C14DM. In mammalian cells and yeast, mitochondria can synthesize phosphatidylglycerol (PG), cardiolipin (CL), phosphatidic acid, and a portion of phosphatidylethanolamine (PE, through the decarboxylation of phosphatidylserine) [19, 60, 61]. Other lipids including phosphatidylcholine, sphingolipids and sterols have to be imported. Changes in mitochondrial lipidome, especially the abundance of CL and PE, can lead to ETC defects [62, 63]. It is therefore possible that defects in sterol synthesis at the ER alter the lipid composition of mitochondrial membrane in *c14dm⁻* mutants. Such membrane modification may negatively affect the functions of ETC including the terminal F0-F1 complex, leading to the accumulation of proton motive force (ΔΨm), leakage of electron and partial reduction of oxygen.

In comparison to WT and add-back parasites, *c14dm⁻* mutants showed significant impairment in oxygen consumption yet produced higher than WT-level of ATP in a glucose-free buffer (Fig 4). Similarly, the cellular ATP content in *c14dm⁻* mutants was 30–40% higher than WT and add-back parasites when cells were incubated in the presence of glucose (Fig 4). These results suggest that without C14DM, *Leishmania* parasites downregulate respiration and

rely more on glycolysis to generate energy. The mechanism of this energetics switch remains to be determined. In many eukaryotes, the heterotrimeric AMP-activated protein kinase (AMPK) complex is a key regulator of cellular energy metabolism [64–66]. Future investigation will examine if AMPK activation in response to C14DM ablation is responsible for reduced respiration and increased glucose consumption via glycolysis.

Consistent with the increased dependency on glucose as an energy source, *c14dm⁻* mutants were extremely sensitive to glucose starvation (Fig 5). This hypersensitivity may arise from the lack of energy production after glucose is depleted via glycolysis and the rapid increase in ROS due to respiration deficiency (Figs 2 and 5). An alternative and not mutually exclusive explanation is that the accumulation of aberrant sterol intermediates in *c14dm⁻* mutants compromised the progression of autophagy, a possibility currently under investigation. Interestingly, supplementation of either glucose or glycerol at concentrations above 100 μM completely rescued the survival of *c14dm⁻* mutants (Fig 5 and S4 Fig). While glycerol could be converted into glucose, it is unlikely to account for the rescue as the energy required for gluconeogenesis exceeds that generated from glycolysis. Instead, it is possible that *Leishmania* parasites metabolize glycerol into glycerol-3-phosphate (by glycerol kinase) and then dihydroxyacetone phosphate (by the mitochondrial FAD-dependent glycerol-3-phosphate dehydrogenase) which can be incorporated into the glycolysis pathway, as described in *T. brucei* [50, 51]. We also observed increased uptake of glucose and glycerol by *c14dm⁻* mutants (Fig 6), a possible adaptive response to their high demand for carbon sources. Future research will focus on effects on sterol synthesis on the permeability of plasma membrane to probe the mechanism of this phenotype.

Next, we examined the involvement of mitochondrial defects in the previously reported heat sensitivity exhibited by *c14dm⁻* mutants. Despite the high mitochondrial ROS, *c14dm⁻* parasites were fully viable at ambient temperature in the presence of glucose or glycerol. However, within 3 hours of exposure at 37 ˚C, the ΔΨm in *c14dm⁻* mutants showed signs of collapse, along with a significant increase in cytosolic ROS level and the release of cytochrome *c* to the cytosol (Fig 7). One possibility is that similar to the effect on plasma membrane, the accumulation of aberrant sterol intermediates compromises the mitochondrial membrane integrity at high temperature in *c14dm⁻* parasites. Although the presence of a canonical apoptotic pathway in *Leishmania* is debatable, earlier studies have suggested the involvement of mitochondrial contents such as cytochrome *c* in causing cell death [32, 67]. In addition, treatment with antioxidant GSH could partially rescue the heat induced cell death *c14dm⁻* mutants, and the ROS generating agent antimycin A made WT *L. major* vulnerable to heat (Fig 7). Together, these findings suggest that ROS accumulation is partially responsible for the heat sensitivity in *c14dm⁻* parasites.

Finally, we also observed heightened sensitivity of *c14dm⁻* to pentamidine, in both the 48-hour growth inhibition assay and the one-hour exposure assay (Fig 8). Previous studies on the amphotericin B resistant lines in *L. mexicana* have identified mutations in sterol biosynthesis and hypersensitivity to pentamidine to be a common feature among these parasites [68, 69]. Our results here are in line with those previous reports as *c14dm⁻* mutants are highly resistant to amphotericin B due to their lack of ergostane-based sterols [7]. It is possible that the elevated ΔΨm and/or mitochondrial ROS cause accumulation of pentamidine in the mitochondria of *c14dm⁻*, making them more susceptible to this drug than WT parasites [55, 56].

In summary, we have shown that deletion or inhibition of C14DM leads to significant alterations in the mitochondria highlighted by increases in ΔΨm and ROS levels, along with deficiency in respiration. These defects make *c14dm⁻* extremely vulnerable to glucose restriction and contribute to their hypersensitivity to heat. Thus, in addition to its stabilizing effects on the plasma membrane, sterol synthesis plays important roles in the regulation of cellular energetics, redox homeostasis and parasite fitness under stress. Analysis of the dynamics of ER-

mitochondria transport, the mitochondrial lipid composition, and the spatial distributions and activities of ETC complexes in *c14dm⁻* would shed light on the mechanistic aspect of how endogenous sterols influence mitochondria in *Leishmania*. Furthermore, the hypersensitivity of *c14dm⁻* to pentamidine raises the possibility of using azole-pentamidine or amphotericin B-pentamidine combinations against leishmaniasis. Future studies may also explore the application of glycolysis inhibitors or ROS inducers in conjugation with sterol biosynthesis inhibitors.

## Materials and methods

### Materials

Tetramethylrhodamine ethyl ester (TMRE), rhodamine 123, Mitotracker CMXRos, MitoSox Red, and dihydroethidium (DHE) were purchased from Thermo Fisher Scientific. Itraconazole (ITZ) was purchased from LKT Laboratories. Antimycin A was from ENZO Life Sciences. The MitoXpress oxygen consumption assay kit was purchased from Luxcel Biosciences. For ATP analysis, the CellTiter-Glo luminescent assay kit was purchased from Promega. All other chemicals were purchased from VWR International or Sigma Aldrich Inc. unless otherwise specified.

### *Leishmania* culture

*L. major* LV39 clone 5 WT (Rho/SU/59/P), *c14dm⁻* (*C14DM*-null mutant), *smt⁻*, *c14dm⁻/ +C14DM* (episomal add-back), and *smt⁻/+SMT80* promastigotes were cultivated at 27 ˚C in complete M199 media (with 10% fetal bovine serum and additional supplements, pH 7.4) [70]. *Leishmania mexicana* (MNYC/BZ/62/M379) axenic amastigotes were cultured using an amastigote medium based on the Drosophila Schneider's medium supplemented with 20% fetal bovine serum and 0.0015% hemin (pH 5.5) in vented flasks in a humidified 32 ˚C/5% $CO_2$ incubator [71]. Culture density and cell viability were determined by hemacytometer counting and flow cytometry after propidium iodide (PI) staining, respectively as previously described [72]. In this study, log phase promastigotes referred to replicative parasites at 2.0–6.0 x $10^6$ cells/ml, and stationary phase promastigotes referred to non-replicative parasites at densities higher than 2.0 x $10^7$ cells/ml.

### Determination of mitochondrial membrane potential (ΔΨm) and reactive oxygen species (ROS) level

ΔΨm was determined by flow cytometry after staining with TMRE or rhodamine 123 as previously described [73]. Briefly, promastigotes were resuspended in complete M199, serum-free M199, phosphate buffered saline (PBS, pH 7.2) containing 5.5 mM of glucose, or PBS alone at 2.0 x $10^6$ cells/ml; TMRE was added to a final concentration of 100 nM; after incubation at 27 ˚C for 15 min, the mean fluorescence intensity (MFI) was determined using an Attune NxT Flow Cytometer. Control samples include WT and *c14dm⁻* parasites pretreated with 75 μM of carbonyl cyanide m-chlorophenyl hydrazone (CCCp) for 15 min and WT parasites grown in the presence of 0.5 μM of ITZ for 48 hours.

Mitochondrial superoxide accumulation was determined as described previously with minor modifications [33]. Promastigotes were resuspended in complete M199, serum-free M199, PBS containing 5.5 mM of glucose, or PBS alone at 1.0 x $10^7$ cells/ml and labeled with 5 μM of MitoSOX red at 27 ˚C. After 25 min, MFI was measured by flow cytometry as described above. To examine the level of cytosolic ROS, parasites were labeled with 5 μM of DHE for 30 min at 27 ˚C followed by flow cytometry analysis. Parasites pretreated with 5 μM

of antimycin A, 75 μM of CCCp, or 100 μM of $H_2O_2$ for 15 min were used in these assays as controls. Such treatments did not affect parasite viability based on PI-staining (<5% PI positive). *C14dm⁻* mutants grown in the presence of 1 mM or 5 mM of glutathione (GSH) for 48 hours were also included. In all flow cytometry experiments, 20,000 events were acquired for each sample.

## Determination of superoxide dismutase (SOD) activity

SOD activity in whole cell extracts was measured as described previously for *L. amazonensis* [74]. Briefly, 2 x $10^8$ promastigotes were harvested, washed twice with PBS and resuspended in a hypotonic buffer (5 mM of Tris–HCl, 0.1 mM of EDTA, and 1 x complete EDTA-free protease inhibitor cocktail, pH 7.8) at a final concentration of 5 x $10^8$ cells/ml. To prepare lysates, cells were subjected to three freeze-thaw cycles alternating between liquid nitrogen and a 37 ˚C water bath. Cell lysis was confirmed by microscopy. Lysates were clarified by centrifugation (12,000 g for 30 min at 4 ˚C) and protein concentrations were determined using a BCA protein assay kit (Thermo Fisher Scientific). SOD activity in whole cell extracts was measured using the EnzyChrome superoxide dismutase assay kit (BioAssay Systems) according to the manufacturer's protocol.

## Digitonin fractionation and Western blot

Crude mitochondria-enriched fractions were generated by digitonin fractionation as described previously [75]. Briefly, log phase promastigotes were incubated for 10 min on ice in 20 mM of Tris-HCl (pH 7.5), 0.6 M of sorbitol, 2 mM of EDTA and 0.025% (w/v) of digitonin at 1 x $10^8$ cells/ml at room temperature. As a positive control for mitochondrial membrane disruption, WT parasites were treated with 0.035% (w/v) of digitonin at 37 ˚C. After centrifugation (8000 g, for 30 min at 4 ˚C), the mitochondria-enriched pellet was separated from the supernatant (enriched for cytosolic proteins). Fractions of equal cell equivalents were subjected to SDS-PAGE and immunoblotting. Primary antibodies included rabbit-anti-*T. brucei* cytochrome *c* (1:500 dilution) [32], rabbit-anti-*L. amazonensis* SODA and SODB (1:500), rabbit-anti-*L. major* ISCL (1:500) [34], and rabbit-anti-*Leishmania* HSP83 (1:5000) antisera [7]. The goat-anti-rabbit IgG conjugated with HRP (1:5000) was used as secondary antibody.

## Measurement of oxygen consumption and ATP production

Oxygen consumption was determined using the MitoXpress Intracellular Oxygen Assay kit (Luxcel Biosciences, Cork, Ireland) as described previously [45]. Briefly, log phase parasites were washed once with PBS and resuspended in DMEM or a mitochondrial respiration buffer (5.5 mM of sodium pyruvate, 5.5 mM of 2-deoxy-D-glucose in HBSS, pH 7.2) at 2.0 x $10^7$ cells/ ml. For each sample, 100 μl of parasite suspension (corresponding to 2.0 x $10^6$ cells) was added in triplicates to a 96 black-well microtiter plate. WT parasites pre-treated with 10 μM of antimycin A or 75 μM of CCCp were used as controls. 10 μl of MitoXpress (1 μM) was added to each well. A layer of mineral oil (100 μl/well) was added over the cells to prevent diffusion of atmospheric oxygen. MitoXpress signals were measured at 90 s intervals for 25–60 min using excitation/emission wavelengths of 380 nm/650 nm in a BioTek synergy 4 fluorescence microplate reader. For experiments in DMEM, we could not continue beyond 25 min when the readings from WT cells reached the saturation point. Wells containing buffer only and buffer + MitoXpress were included as blanks. For each cell line, the relative OCR was calculated by subtracting the 0 min reading from the 15 min reading.

To measure ATP production, log phase parasites were washed once with PBS and resuspended in an assay buffer containing 21 mM of HEPES, 0.7 mM of $Na_2HPO4$, and 137 mM of

NaCl (pH 7.2) in the presence or absence of glucose (5.5 mM), sodium pyruvate (5.5 mM), 2-deoxy-D-glucose (5.5 mM) or sodium azide (20 mM) at 4.0 x 10^6 cells/ml. After incubation at 27 ˚C for one hour, 25 µl of suspension (corresponding to 1.0 x 10^5 cells) were transferred to a 96 black-well plate, mixed with equal volume of the CellTiter-Glo reagent and incubated for 10 min in the dark. Bioluminescence was measured using a DTX880 microplate reader as described [23]. For each experiment, an ATP standard curve was prepared by diluting pure ATP stock in the assay buffer.

## Microscopy

To stain mitochondria, live parasites were resuspended in DMEM at $1.0 \times 10^6$ cells/ml and labeled with 100 nM of Mitotracker CMXRos or TMRE. After incubation at 27 ˚C (30 min for Mitotracker and 15 min for TRME), parasites were washed once with PBS, fixed and processed for fluorescence microscopy as previously described [15]. MitoSox Red labeling (5 µM for 25 min at 27 ˚C) of live parasites was performed similarly without fixation using an Olympus BX51 Upright Fluorescence Microscope. DNA was stained with DAPI (2.5 µg/ml). Transmission electron microscopy of log and stationary phase promastigotes was performed as previously described [76].

## Starvation and heat stress assays

To measure heat sensitivity, log phase promastigotes were incubated in complete M199 media for 0–24 h at either 27 ˚C or 37 ˚C/5% $CO_2$. Effects of antimycin A (10 µM) and GSH (1 mM or 5 mM) on heat sensitivity were also monitored. For starvation response, log phase promastigotes were washed once with HBSS and resuspended in HBSS supplemented with glucose (5.5 mM), 2-deoxy-D-glucose (5.5 mM), aspartate (5.5 mM), sodium pyruvate (5.5 mM), or glycerol (0.1 mM) at 2.5 x 10^6 cells/ml and incubated for 0–24 h at 27 ˚C. In these stress assays, culture densities and percentages of dead cells were determined over time as previously described [72].

## Glucose and glycerol uptake assays

Uptake of glucose and glycerol were monitored as previously described with minor modifications [53, 77]. For glucose incorporation, log phase promastigotes were washed twice with PBS and resuspended at 5.0 x 10^7 cells/ml in PBS in the presence of 100 µM of D-[U-^14C] glucose (250 mCi/mmol, PerkinElmer Inc.). The reaction mixture (200 µl each in triplicates) was immediately loaded onto 100 µl of dibutyl phthalate in 1.5 ml Eppendorf tubes. Incubation was performed at room temperature and terminated at the indicated times by pelleting cells through the oil layer through centrifugation (16,000 g, 1 min). Tubes were then flash frozen with liquid nitrogen and the bottoms (containing cell pellets) were clipped off and submerged in scintillation fluid containing 1% SDS for 24 h, followed by scintillation counting. To measure glycerol uptake, log phase promastigotes were washed twice in a transport buffer (33 mM of HEPES, 98 mM of NaCl, 4.6 mM of KCl, 0.55 mM of $CaCl_2$, 0.07 mM of $MgSO_4$, 5.8 mM of $Na_2PO4$, 0.3 mM of $NaHCO_3$, and 14 mM of glucose, pH 7.3) and resuspended at 1.0 x 10^8 cells/ml in the same buffer. Uptake was measured at room temperature by mixing 100 µl of cells with 100 µl of transport buffer containing 8 µM of [^14C] glycerol (0.125 µCi each in triplicates, PerkinElmer Inc.). Incorporation of radioactivity was determined as described above.

## Susceptibility to pentamidine

To determine the effective concentration of pentamidine to inhibit parasite growth by 90%, 50% and 25% (EC90, EC50 and EC25), log phase promastigotes were inoculated in complete

M199 media at $2.0 \times 10^5$ cells/ml in 0–25 μM of pentamidine. Culture densities were measured after 48 hours of incubation in 24-well plates. EC90, EC50 and EC25 were determined using cells grown in the absence of pentamidine as controls. To assess the leishmanicidal activity of pentamidine in short time exposure, log phase promastigotes were inoculated in complete M199 media containing 0, 2.5 μM, or 25 μM of pentamidine. For *c14dm⁻*, half of them were also treated with 75 μM of CCCp. After one hour, cells were washed twice with PBS and re-inoculated in complete M199 media at $2.0 \times 10^5$ cells/ml. Culture densities were then determined daily for six days.

## Statistical analysis

Unless specified otherwise, assay values in all figures are averaged from three to five independent repeats and error bars represent standard deviations. Differences were assessed by the Student's t test (for two groups) or one-way ANOVA (for three or more groups) using the Sigmaplot 13.0 software (Systat Software Inc, San Jose, CA). *P* values indicating statistical significance were grouped in all figures (***: $p < 0.001$, **: $p < 0.01$, *: $p < 0.05$).

## Supporting information

**S1 Fig. *C14dm⁻* promastigotes show increased mitochondrial surface area but maintain mitochondrial membrane integrity at ambient temperature.** (A) After Mitotracker staining (Fig 1A), the average mitochondrial surface areas in log phase promastigotes of WT, *c14dm⁻*, and *c14dm⁻/+C14DM* were determined using Image J (~200 cells were analyzed for each parasite line, au: arbitrary unit). (B) Log phase promastigotes were lysed with digitonin at 37 ˚C or room temperature (RT) and mitochondria enriched membrane fractions (Mem.) were separated from cytosolic fractions (Sup.) as described in Materials and Methods. Western blots were performed using antibodies against cytochrome *c*, ISCL (a mitochondrial membrane protein), and HSP83 (a cytosolic protein).
(PDF)

**S2 Fig. *C14dm⁻* and *smt⁻* mutants show increased ΔΨm and mitochondrial ROS.** Log phase and stationary phase (day 1-day 4) promastigotes were resuspended in PBS and labeled with 5 μg/ml of rhodamine 123 for 15 min (**A**) or with 5 μM of MitoSox Red for 25 min (**B**) at room temperature. MFIs were determined by flow cytometry. Error bars represent standard deviations from three independent experiments.
(PDF)

**S3 Fig. *C14dm⁻* mutants accumulate ROS in the mitochondria.** Log phase promastigotes were labeled with 5 μM of MitoSox Red for 25 min (**A**) or 5 μM of DHE for 30 min (**B**) and MFIs were determined by flow cytometry. Effects of AA (5 μM) and $H_2O_2$ (100 μM) on WT parasites were also monitored. Error bars represent standard deviations from three experiments.
(PDF)

**S4 Fig. 100 μM of glucose or glycerol is sufficient to rescue *c14dm⁻* mutants in PBS.** Log phase promastigotes were incubated in PBS in the absence or presence of glucose (100 μM) or glycerol (100 μM) and percentages of dead cells were determined by flow cytometry at the indicated time points. Error bars represent standard deviations from three experiments.
(PDF)

**S5 Fig. ITZ-treated WT parasites are hypersensitive to glucose depletion.** *L. major* WT promastigotes were cultivated in solvent alone (0.1% w/v of DMSO) or 0.2 μM of ITZ for 48

hours. Cells were then incubated in HBSS and percentages of dead cells were determined by flow cytometry at the indicated time points. Error bars represent standard deviations from three experiments.
(PDF)

**S6 Fig. *C14dm⁻* mutants rescued by glucose or glycerol show less severe mitochondrial defects.** Log phase promastigotes were incubated in PBS in the absence or presence of glucose (100 μM) or glycerol (100 μM) for 24 hours. ΔΨm (**A**) and mitochondrial ROS level (**B**) were determined by flow cytometry. Control cells represent cells analyzed at the beginning of incubation. †: no viable *c14dm⁻* mutants were available after 24 hours. Error bars represent standard deviations from three experiments.
(PDF)

# Acknowledgments

We thank Dr. Norma Andrews (University of Maryland College Park) for providing us the rabbit anti-*L. amazonensis* SODA and SODB antisera, Dr. André Schneider (University of Bern) for the kind gift of rabbit anti-*T. brucei* cytochrome *c* antibody, Dr. Wandy Beatty (Washington University in St. Louis) for transmission electron microscopy, and Dr. Catherine Wakeman (Texas Tech University) for usage on the BioTek synergy 4 fluorescence microplate reader.

# Author Contributions

**Data curation:** Sumit Mukherjee, Samrat Moitra, Wei Xu, Veronica Hernandez.

**Formal analysis:** Sumit Mukherjee, Samrat Moitra, Veronica Hernandez, Kai Zhang.

**Funding acquisition:** Kai Zhang.

**Investigation:** Sumit Mukherjee, Kai Zhang.

**Methodology:** Sumit Mukherjee, Samrat Moitra, Wei Xu, Veronica Hernandez.

**Writing – original draft:** Sumit Mukherjee, Samrat Moitra, Kai Zhang.

**Writing – review & editing:** Sumit Mukherjee, Samrat Moitra, Kai Zhang.

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
