## [Decision Letter · Decision Letter 0]

4 Mar 2020

Dear Dr Zhang,

Thank you very much for submitting your manuscript "Sterol 14alpha-demethylase is vital for mitochondrial functions and stress tolerance in Leishmania major" for consideration at PLOS Pathogens. As with all papers reviewed by the journal, your manuscript was reviewed by members of the editorial board and by several independent reviewers. In light of the reviews (below this email), we would like to invite the resubmission of a significantly-revised version that takes into account the reviewers' comments.

All referees considered this to be a solid piece of experimental work. However, views on acceptability for publication in Plos Pathogens were divided. Reviewer 1, in particular, felt that the gain in knowledge was incremental given that others have already identified roles of sterols in mitochondrial membrane composition and the current work is an additional piece of phenotypic analysis of the mutant line first published in Plos Pathogens in 2014. However, as I have an interest in this field, I was particularly taken with the observation that the C14DM mutant has a substantially increased mitochondrial membrane potential. We have developed a series of amphotericin B resistant lines (L. mexicana) (PMID: 30716073 & PMID: 28622334). They have mutations in different sterol pathway enzymes, converging on a loss of ergostane type sterols. We find hypersenstivity to pentamidine a common feature in these parasites (and others unpublished). Since pentamidine's activity has some dependence on mitochondrial membrane potential and accumulation into that organelle (PMID: 12435669), the findings here offer a potentially satisfying explanation to this observation. You have previously shown the mutant is resistant to Amphotericin B. It would be useful to include data here on sensitivity to pentamidine to which they are likely to be hypersensitive). This would be a simple experiment.

It would be further interesting to see if you could work out how long parasites need to be exposed to pentamidine (in wash out experiments) before they commit to death. If, as is true for African trypanosomes, fairly short exposures are needed (does dependent) it would be possible to set up experiments to see if transiently diminishing mitochondrial potential with CCCP could alter time dependence of pentamidine exposure to commit to death. Similar work was done with T. brucei as a possible marker on how to design experiments (PMID: 8103625).

If you can place your work in a more translational context of chemotherapeutic relevance I feel the work's significance will be greatly enhanced.

A few additional comments I'd make, in addition to those of the other reviewers:

Line 14 - mitochondrial not mitochondria

Line 68 - can a comment on amphotericin B activity in amastigotes be included given their apparent low level of ergostanes?

Line 109 "quite unique" is not really correct, either in the meaning of the word "unique" not in biological relevance. Lots of organisms have a single mitochondrion, so just drop the "quite unique"

Line 140: add "at least with regard to cytochrome c localisation between "integrity" and "under" (other aspects of the membrane's integrity weren't looked at)

Line 327: also in considering more pharmacological relevance in a revised manuscript - could triazole/pentamidine combinations be considered (assuming you find pentamidine hypersensitivity)

REFERENCES: Please check the format of each against Journal requirements. Note that reference 2. uses abbreviated journal title name, others use whole title, so check for consistency.

We cannot make any decision about publication until we have seen the revised manuscript and your response to the reviewers' comments. Your revised manuscript is also likely to be sent to reviewers for further evaluation.

Sincerely,

Michael P. Barrett, PhD

Guest Editor

PLOS Pathogens

David Horn

Section Editor

PLOS Pathogens

Kasturi Haldar

Editor-in-Chief

PLOS Pathogens

orcid.org/0000-0001-5065-158X

Michael Malim

Editor-in-Chief

PLOS Pathogens

orcid.org/0000-0002-7699-2064

All referees considered this to be a solid piece of experimental work. However, views on acceptability for publication in Plos Pathogens were divided. Reviewer 1, in particular, felt that the gain in knowledge was incremental given that others have already identified roles of sterols in mitochondrial membrane composition and the current work is an additional piece of phenotypic analysis of the mutant line first published in Plos Pathogens in 2014. However, as I have an interest in this field I was particularly taken with the observation that the C14DM mutant has a substantially increased mitochondrial membrane potential. We have developed a series of amphotericin B resistant lines (L. mexicana) (PMID: 30716073 & PMID: 28622334). They have mutations in different sterol pathway enzymes, converging on a loss of ergostane type sterols. We find hypersenstivity to pentamidine a common feature in these parasites (and others unpublished). Since pentamidine's activity has some dependence on mitochondrial membrane potential and accumulation into that organelle (PMID: 12435669), the findings here offer a potentially satisfying explanation to this observation. You have previously shown the mutant is resistant to Amphotericin B. It would be useful to include data here on sensitivity to pentamidine to which they are likely to be hypersensitive). This would be a simple experiment.

It would be further interesting to see if you could work out how long parasites need to be exposed to pentamidine (in wash out experiments) before they commit to death. If, as is true for African trypanosomes, fairly short exposures are needed (does dependent) it would be possible to set up experiments to see if transiently diminishing mitochondrial potential with CCCP could alter time dependence of pentamidine exposure to commit to death. Similar work was done with T. brucei as a possible marker on how to design experiments (PMID: 8103625).

If you can place your work in a more translational context of chemotherapeutic relevance I feel the work's significance will be greatly enhanced.

A few additional comments I'd make, in addition to those of the other reviewers:

Line 14 - mitochondrial not mitochondria

Line 68 - can a comment on amphotericin B activity in amastigotes be included given their apparent low level of ergostanes?

Line 109 "quite unique" is not really corerct, either in the meaning of the word "unique" not in biological relevance. Lots of organisms have a single mitochondrion, so just drop the "quite unique"

Line 140: add "at least with regard to cytochrome c localisation between "integrity" and "under" (other aspects of the membrane's integrity weren't looked at)

Line 327: also in considering more pharmacological relevance in a revised manuscript - could triazole/pentamidine combinations be considered (assuming you find pentamidine hypersensitivity)

REFERENCES: Please check the format of each against Journal requirements. Note that reference 2. uses abbreviated journal title name, others use whole title, so check for consistency.

Reviewer's Responses to Questions

**Part I - Summary**

Reviewer #1: This article by Mukherjee et al. investigates the role of the sterol biosynthetic enzyme C14-�-demethylase (C14DM) in mitochondrial function in promastigotes of Leishmania major. This enzyme is of interest, because it is the target of azole compounds that have efficacy as antifungals and have been investigated as potential anti-leishmanial drugs. In previous studies, the authors have generated the c14dm- null mutant and shown that plasma membrane fluidity is decreased in this mutant, that it is susceptible to increased temperature, and that its virulence in mice is greatly reduced. Of interest, these effects are probably due to accumulation of C14-methylated sterol precursors when the demethylase is knocked out or inhibited rather than to the absence of the mature sterol. The current study focuses on the effects of the c14dm- mutation, and of treatment with the C14DM inhibitor itraconazole (ITZ), on the mitochondrion. Although most sterol end products in eukaryotic cells (e.g., cholesterol or ergosterol) are present in the plasma membrane or vesicles that traffic to the PM, sterols are also present in organellar membranes including the mitochondrion, albeit at lower levels.

Indeed, the authors find major alterations in promastigote mitochondria in the c14dm- mutant and in parasites treated with ITZ. Mitochondria increase in size, the mitochondrial membrane potential is significantly increased, and levels of reactive oxygen species (ROS) increase by 10-15-fold in the mitochondrion, probably due to leakage of electrons from the mitochondrial electron transport chain due to alterations in the inner mitochondrial membrane. Furthermore, oxygen consumption is reduced in the c14dm- mutants, and the parasites appear to switch to greater reliance upon glycolysis for generation of cellular ATP. The promastigotes thus become highly sensitive to glucose restriction, but supplementation of the medium with either glucose or glycerol protects the parasites against loss of viability. Furthermore, shifting temperature from 27° to 37° also induces mitochondrial alterations in these mutants, including reduced membrane potential, leakage of cytochrome oxidase out of the mitochondrion, and increased cytosolic ROS, leading to cell death. Overall, these studies implicate a critical role for sterols in maintenance of mitochondrial function in these parasites and imply that azoles may function, at least in part, by impairing mitochondrial function.

This is a well-executed study that employs a variety of assays to assess mitochondrial function in wild type, c14dm- mutant, and ITZ-treated promastigotes. Appropriate scientific rigor is applied, with multiple replicates and statistical analysis of data. The paper reports a substantial body of work, and the studies should be of interest to the molecular parasitology community. The major new material is the elucidation of effects of blocked sterol biosynthesis on mitochondria, as opposed to earlier studies by this group that focused upon the plasma membrane, the principal site of sterol accumulation. Nonetheless, the novelty is probably not exceptional. The c14dm- mutant has been investigated previously and shown to have significant deficiencies, including on virulence. Sterols have been known to partition into intracellular organelles for some time, as indicated in various review articles published over more than a decade. Previous studies on trypanosomatids, reviewed by these authors on pages 5-6 of this manuscript, have implicated the mitochondrion in impairments induced by azoles, although they have not been as detailed or definitive as the current work. Hence, this is a strong study focused on a pathway of interest for possible drug development, but perhaps not of exceptional novelty or interest.

Reviewer #2: In this manuscript, Mukherjee et al. have explored the impact of genetic or pharmacological ablation of sterol 14�-demethylase (c14dm-) activity on the function and integrity of the mitochondrion in Leishmania major. Previous studies have demonstrated a link between disruption of sterol biosynthesis and the mitochondrion in both Leishmania and Trypanosoma, however, these studies were observational and lacked the systematic evaluation offered in this manuscript. The authors, through a meticulous set of experiments, that were appropriately controlled and which offered clear outcomes to testable hypotheses, have demonstrated that disruption of C14DM leads to the interruption of energy metabolism in the mitochondrion, making these cells highly dependent on glycolysis to meet their energy needs. The dysfunction in the mitochondrion appears to be linked to an impairment in ATP synthesis via ATP synthase, which leads to an increase in mitochondrial membrane potential and ROS production – the likely cause of the increased sensitivity of these c14dm- cells to stress.

The quality of the work in this paper is exceptionally high, the results unambiguous, and the work reaffirms C14DM, and indeed the sterol biosynthetic pathway, as an important drug target in these parasites worthy of further exploration. The work expands our current knowledge of the mechanisms of action of the azole drugs and, since it is highly likely it can be extrapolated to other trypanosomatids, should be of broad interest to the field. I have no suggested major revisions, and only a few minor comments to improve the overall manuscript.

Reviewer #3: This manuscript builds on the authors’ prior work on endogenous sterol synthesis in Leishmania major and the implications for drug design. The manuscript is clearly written and cites the existing literature well. A major strength of work from this group (including this manuscript) is the use of genetic knockout mutants (c14dm-/- and smt-/-) to understand parasite biology instead of the sole use of chemical inhibitors of these enzymes that have potential for off target impacts. The authors state that further characterization of sterol synthesis in the clinically relevant stages of Leishmania would provide interesting and potentially different insights. It is a point well made by the authors but may complicate some of the stated significance (Line 19). Some of the conclusions with regard to “adaptive” responses are overstated and not in line with the data.

**Part II – Major Issues: Key Experiments Required for Acceptance**

Reviewer #1: (No Response)

Reviewer #2: None suggested

Reviewer #3: As written, some of the conclusions regarding the relative roles of glycolysis and mitochondrial respiration in the c14dm-/- mutants vs WT parasites are a bit misleading (Fig 4).

Figure 4A: The graph shows that the c14dm-/- mutant exhibits reduced mitochondrial respiration as compared to the WT and add-back parasites. But what is presented are not the basal respiration rates for each line and therefore not a fair initial comparison. In this experiment, OCR was measured in the presence of 2-DG (line 478) to block glucose utilization – and because there is a bigger effect on the mutant, it appears as if respiration in the mutant line is impaired under basal conditions, which may or may not be the case. Therefore, a figure should be included to show basal respiration rate (OCR) measured in log-phase c14dm-/- mutants as compared to WT and add-back parasites incubated with a full complement of substrates. This data would determine if the mitochondrial defects observed (e.g. Fig1/2) result in defects in respiration. Based on the buffer composition described for Fig. 4 the OCR data appear to be oxygen consumption solely from pyruvate. These conditions, and the inferences gained need to be clarified in the text.

The authors should include the raw MitoXpress data for these experiments - linear plots as supplemental data. Also, the OCR value should be normalized to parasite number or protein in each sample.

Secondly, based on results presented in Fig 4, the authors emphasize increased dependence of the c14dm-/- mutant on glycolysis which might be expected because of the ‘mitochondrial defects’ observed in this mutant. However, as shown in Fig. 4B, most of the ATP production in the c14dm-/- mutant comes from mitochondrial respiration, not glycolysis. While glycolysis might contribute a bit more to the ATP produced by the mutant than the WT, it appears to be a minor contribution. Moreover, the authors do not include data to confirm the presumptive contribution of glycolysis to the residual ATP produced by the mutant in the presence of sodium azide. Data should be included to show that this signal is completely ablated if an excess of 2-DG is added to block glycolysis.

It is unclear why the buffer used in Fig. 4B (line 486) contains 2-DG. The figure legend does not state that 2-DG is in the buffer and may confuse readers. How is ATP production is expected to be altered in the absence of 2-DG in the buffer?

Also, in Fig 4B, in the absence of an exogenous carbon source (for 1 hr), there was no difference in the amount of ATP produced by the c14dm-/- mutants, WT and add-back. What is the major contributor to this basal ATP production – glycolysis, mitochondrial respiration? Does sodium azide or 2-DG treatment differentially impact the mutants and WT parasites? What does the pattern look like when parasites have access to a mixture of usable carbons (and absence of 2-DG in the buffer)? This information should be included and results interpreted in context.

When glucose is the only available exogenous substrate, the c14dm-/- parasites produce more ATP than WT parasites (there is no significant difference between mutant and add-back parasites) and the mutants take up more glucose and glycerol than the WT or add-back parasites. These data support the idea that the mutants might rely more heavily on glucose than other substrates to fuel energy production. Exogenous pyruvate or aspartate fail to rescue mutants from glucose starvation. In this scenario, is it possible to rule out impaired plasma membrane or mitochondrial permeability to other potential TCA substrates in the c14dm-/- mutants?

Along the same line, in some experiments, pyruvate is included in the parasite incubation buffer – presumably to fuel the TCA cycle and mitochondrial respiration. It doesn't seem like the inclusion of pyruvate makes a difference in the ability of WT parasites to generate ATP. The amount of ATP produced by all parasite lines in HBSS alone (Fig 4B, first set of bars) and HBSS + pyruvate present (Fig. 4C, first set of bars) appears to be similar. Why doesn’t ATP production increase in the presence of exogenous pyruvate and what is the effect on OCR?

At 4 hr in the absence of glucose (Fig4C) – but in the presence of pyruvate - the c14dm-/- mutants start to lose the capacity for ATP synthesis. Given that mitochondrial respiration is the main source of ATP in the c14dm-/- mutants (Fig. 4B), exogenous pyruvate should sustain ATP production at a higher level in the mutant if it is being utilized. Is ATP production lost at a higher rate in the mutant in the absence of exogenous pyruvate?

Has it been demonstrated that pyruvate is transported across the parasite plasma membrane so that it can be utilized by the mitochondria? In order to draw conclusions about substrate utilization and mitochondrial function, it is important to demonstrate that these important assumptions of the assay (e.g. substrate transport over the plasma membrane) are verified. Some of these data can be explain through substrate permeability differences between parasites.

**Part III – Minor Issues: Editorial and Data Presentation Modifications**

Reviewer #1: Minor Points.

I would recommend changing the order of Figs. 5A and 5B, as the latter panel is introduced first in the manuscript.

Reviewer #2: • The error bars on Fig. 4B are very faint and need to be made more visible.

• The method used for assessing cell death by propidium iodide staining, though standard, is not detailed or referenced in the methods.

• There are few grammatical/typographical errors throughout the manuscript that need to be addressed prior to publication.

• Not a revision, but a question for the authors consideration. I’m curious if the increased uptake of glycerol is related to the enhanced activity of AQP1 and if so, how that might affect the sensitivity of the c14dm- cells to antimony drugs. Is something that is known or that has been tested?

Reviewer #3: Line 66-68. What is the evidence in [7] that scavenging predominates and there is a downregulation of ergosterol synthesis? If this is true what is the cause of the attenuated growth in vivo of the mutants (i.e. would there be a reduction in 14-methylated sterols?)? In ref 7 the statement “Clearly, some of the cholesterol was not directly associated with amastigotes instead from mouse cells” suggests that the extent/importance of cholesterol scavenging is not entirely settled with regard to amastigotes.

Fig 3B and line 210-211. Without a loading control or quantification the claim of “similar” seems to be a fair characterization but “slightly higher” may be a stretch.

Line 216: The statement “..outside of SOD deficiency.” might be interpreted, as SOD is sufficient to control ROS (which the data suggest it may not be). Would it be fair to state that parasite SODs appear unaffected but are apparently unable to alleviate excess ROS production?

Line 447 “Such treatments did not affect parasite viability based on PI-staining.” It is unclear if PI was used for all of the flow experiments to verify the integrity of the parasites. If this is the case it would be worth mentioning in the results section to support the conclusions when treatments like CCCP lead to a negative result (i.e. Fig 2A/B). This would further strengthen the authors conclusions as opposed to any results being due to alterations of general viability. Additionally (but unlikely under the stated treatment times) parasite death leading to lysis will not lead to a detectable event (PI pos or negative) to establish viability. Am I correct in assuming that parasites are not PI positive and the number of events (i.e. parasites) is consistent between treatment groups?

A major contribution of the generation of the smt-/- line (used in the 2018 publication) is the ability to compare phenotypes to the c14dm-/- line. This comparison (as noted by the authors) can help delineate if a given phenotype is caused by generation of 14-methylated sterols or absence on an endogenously synthesized sterol. In Figure 5A the smt-/- is used to support the notion that 14-methylated sterols contribute to the dependence on glucose. However, to support the notion that loss of mitochondrial membrane potential in the c14dm-/- is the cause of this glucose dependence it would be preferrable to see a side by side TMRE values (i.e. smt-/- and c14dm-/- ).

Lines 196-199 reference the 2018 publication as “not as profound” but a direct comparison/fold change would be more illustrative.

PLOS authors have the option to publish the peer review history of their article (what does this mean?). If published, this will include your full peer review and any attached files.

Reviewer #1: No

Reviewer #2: No

Reviewer #3: No
---

## [Decision Letter · Decision Letter 1]

14 Jul 2020

Dear Dr. Zhang,

We are pleased to inform you that your manuscript 'Sterol 14-alpha-demethylase is vital for mitochondrial functions and stress tolerance in Leishmania major' has been provisionally accepted for publication in PLOS Pathogens.

Please also address Reviewer #1 minor suggestions for corrections in the text (see below).

Best regards,

Michael Barrett

Guest Editor

PLOS Pathogens

David Horn

Section Editor

PLOS Pathogens

Kasturi Haldar

Editor-in-Chief

PLOS Pathogens

orcid.org/0000-0001-5065-158X

Michael Malim

Editor-in-Chief

PLOS Pathogens

orcid.org/0000-0002-7699-2064

The authors have made substantial changes to the manuscript including new experiments that have added to the potential translational impact of the work as well as responding to each of the queries raised by referees and addressing those where appropriate. All in all the work is good and represents an important breakthrough in our understanding of the roles played by sterols in Leishmania biology and how this can impact on chemotherapy.

Reviewer Comments (if any, and for reference):

Reviewer's Responses to Questions

**Part I - Summary**

Reviewer #1: This revised manuscript describes the critical role of the sterol biosynthetic enzyme C14DM in mitochondrial function in Leishmania major. The authors have addressed the comments of the 3 reviewers as well as suggestions made by the Associate Editor. Their responses have been effective and have enhanced the significance of the study. In particular, addressing the increased sensitivity of c14dm null mutant to the clinically employed drug pentamidine underscores the significance of this basic genetic study to drug treatment. The observation that either c14dm null mutants or treatment with inhibitory triazole drugs increases the efficacy of pentamidine raises the potential for combination drug therapy that would improve regimens involving delivery of pentamidine.

While this reviewer was somewhat concerned about the novelty of the original submission, this issue is addressed both by new experiments and by the recognition that the current study does indeed highlight in new ways the role of sterols in the mitochondrial function of these parasites. Thus inhibition of sterol biosynthesis has traditionally focused on the effects on the plasma membrane, where the majority of sterols reside. The current detailed study shows that there are important biological consequences to either inhibiting this enzyme or knocking out the C14DM gene that operate at the level of the mitochondrion rather than the plasma membrane. It is known from previous work on various systems that sterols are present in the inner mitochondrial membrane and that compounds that inhibit sterol biosynthesis have morphological and biochemical effects on mitochondria. However the current work provides much more mechanistic detail than was available before, showing an increase in the mitochondrial membrane potential and in mitochondrial ROS and increased sensitivity to stresses such as nutrient deprivation and heat following deletion of the C14DM gene. These detrimental effects on the mitochondrion are relevant to pharmacological properties of C14DM inhibitors, and the resulting increased sensitivity to pentamidine is of added pharmacological importance.

**Part II – Major Issues: Key Experiments Required for Acceptance**

Reviewer #1: None suggested.

**Part III – Minor Issues: Editorial and Data Presentation Modifications**

Reviewer #1: I have a few minor suggestions for corrections in the text.

1. Line 69 – Change ‘residue’ to ‘residual’.

2. Line 108 – Change ‘play’ to ‘plays’.

3. Change – ‘2-deoxy-G-glucose’ to ‘2-deoxy-D-glucose’ in lines 228, 250, 272.

4. Line 359 – Change ‘folds’ to ‘fold’.

5. Line 392 – Change ‘Mechanism’ to ‘The mechanism’.

PLOS authors have the option to publish the peer review history of their article (what does this mean?). If published, this will include your full peer review and any attached files.

Reviewer #1: No

---

## [Editor Report · Acceptance letter]

4 Aug 2020

Dear Dr. Zhang,

We are delighted to inform you that your manuscript, "Sterol 14-alpha-demethylase is vital for mitochondrial functions and stress tolerance in Leishmania major," has been formally accepted for publication in PLOS Pathogens.

Best regards,

Kasturi Haldar

Editor-in-Chief

PLOS Pathogens

orcid.org/0000-0001-5065-158X

Michael Malim

Editor-in-Chief

PLOS Pathogens

orcid.org/0000-0002-7699-2064